# Correlated evolution between repertoire size and song plasticity predicts that sexual selection on song promotes open-ended learning

Cristina M Robinson†, Kate T Snyder†, Nicole Creanza*

Department of Biological Sciences, Vanderbilt University, Nashville, United States

**Abstract** Some oscine songbird species modify their songs throughout their lives ('adult song plasticity' or 'open-ended learning'), while others crystallize their songs around sexual maturity. It remains unknown whether the strength of sexual selection on song characteristics, such as repertoire size, affects adult song plasticity, or whether adult song plasticity affects song evolution. Here, we compiled data about song plasticity, song characteristics, and mating system and then examined evolutionary interactions between these traits. Across 67 species, we found that lineages with adult song plasticity show directional evolution toward increased syllable and song repertoires, while several other song characteristics evolved faster, but in a non-directional manner. Song plasticity appears to drive bi-directional transitions between monogamous and polygynous social mating systems. Notably, our analysis of correlated evolution suggests that extreme syllable and song repertoire sizes drive the evolution of adult song plasticity or stability, providing novel evidence that sexual selection may indirectly influence open- versus closed-ended learning.
DOI: https://doi.org/10.7554/eLife.44454.001

**\*For correspondence:**
nicole.creanza@vanderbilt.edu

†These authors contributed equally to this work

**Competing interests:** The authors declare that no competing interests exist.

## Introduction

Song is a learned behavior with a complex evolutionary history in the oscine songbirds. Birds' songs have multiple functions, including species recognition, territory defense, and mate attraction (*Catchpole and Slater, 2003*). All species studied in this expansive clade have certain life stages during which they are more likely to learn and acquire songs, termed sensitive periods (*Brenowitz and Beecher, 2005*; *Marler, 1990*; *Marler and Peters, 1987*; *Rauschecker and Marler, 1987*). In some species, learning is restricted to a short sensitive period early in life, also called a 'critical period' (e. g. ~day 25–90 in zebra finches), after which no new song elements are acquired (*Böhner, 1990*; *Immelman, 1969*; *Nottebohm, 1984*). Other species appear to delay song crystallization until some time in adulthood (*Dowsett-Lemaire, 1979*; *Kipper and Kiefer, 2010*; *Martens and Kessler, 2000*); for example, chipping sparrows appear to have a second sensitive period immediately after their first migration, following which their song crystallizes (*Liu and Kroodsma, 2006*; *Liu and Nottebohm, 2007*). Still other species can continue to acquire new syllables or songs throughout their lives (*Adret-Hausberger et al., 2010*; *Espmark and Lampe, 1993*; *Gil et al., 2001*; *Hausberger et al., 1991*; *Mountjoy and Lemon, 1995*; *Price and Yuan, 2011*). Typically, this spectrum of variation in the timing of the sensitive period is simplified into a dichotomy of 'open-ended learning' and 'closed-ended learning.' While these temporal differences in the song-learning window have been studied for decades, it is unknown how they interact with the evolution of song itself. Previous hypotheses have suggested that seasonal factors, such as environmental variation and breeding season length, play a role in shaping adult song learning (*Nottebohm et al., 1986*; *Smith et al., 1997*; *Tramontin et al., 2001*).

**eLife digest** Every morning, people all over the world are greeted by the sound of songbirds singing to attract mates and defend their homes. Each type of songbird has its own unique song that it learns from other birds in the same species. Maintaining these signature songs is important, because songbirds that sing the wrong tune are unlikely to succeed in breeding. This type of vocal learning is rare in the animal kingdom and is similar to how humans learn to speak.

The time it takes for songbirds to learn their unique song varies between species: some species pick up their song within a few months after hatching, while others may continue learning for years or even the rest of their lives. It remains unclear, however, why certain species spend a longer time learning, and how changing this length of time affects the song they sing. One possibility is that differences in the amount of time spent learning evolved from female songbirds preferring mates with more impressive songs.

To address this possibility, Robinson et al. used previously published data to compare the song characteristics, learning periods, and mating strategies of 67 songbird species. This confirmed that longer learning windows allowed songbirds to develop a larger repertoire of songs with more unique sounds. Robinson et al. also found songbirds that learn elaborate songs over a short period of time quickly evolve either simpler songs or longer phases of learning. Meanwhile, songbirds that learn simple songs over a longer period tended to evolve more elaborate songs or shorten the amount of time they spent learning. Furthermore, species with longer learning were more likely to switch between having one mating partner and multiple partners over several generations.

These findings suggest that by choosing mates with elaborate songs and larger repertoires, female songbirds can end up favoring the evolution of longer learning. Studying the conditions that led to longer or shorter learning in songbirds could help scientists understand why some things, like human language, are easier to learn early in life.

DOI: https://doi.org/10.7554/eLife.44454.002

However, evidence from a small-scale comparative analysis suggests that a longer learning window in a species may be associated with larger average syllable repertoire sizes (*Creanza et al., 2016*).

Birdsong is composed of both culturally and genetically inherited features, any of which may be subject to evolutionary pressures. Two key modes of selection on song might act in conjunction: female choice can favor certain song characteristics, such as superior repertoire size, learning quality, or song performance (*Beecher and Brenowitz, 2005*; *Gil and Gahr, 2002*; *Searcy and Marler, 1984*; *Searcy and Andersson, 1986*), while the inherent metabolic cost of neuroplasticity should theoretically favor a shorter song-learning window and thus reduce the a for a bird to alter its song in adulthood (*Garamszegi and Eens, 2004*; *Nottebohm et al., 1986*; *Tramontin and Brenowitz, 2000*). Therefore, while learning in adulthood or elongated sensitive periods have not been shown to be directly under positive selection (sexual or natural) or to play an explicit role in female preferences, sexual selection acting on certain song features could indirectly favor longer or shorter song-learning windows. However, this theorized connection between sexual selection and adult learning hinges on establishing the evolutionary relationship between song as the target of sexual selection and the neurobehavioral phenotype of song learning, which has not yet been done.

Furthermore, sexual selection is hypothesized to be amplified in species with polygynous social mating systems or high rates of extra-pair paternity (EPP). A recent large-scale study found that polygyny drives faster, but non-directional, evolution of syllable repertoire size, and that syllable repertoire size is negatively correlated with the rate of EPP (*Snyder and Creanza, 2019*). Because of the evolutionary links between these mating strategies and song features, higher rates of EPP and polygyny could potentially have an effect on learning windows. We therefore investigate the evolutionary relationship between the critical learning period and mating strategies.

Here, we take a comparative, computational approach to the evolutionary history of open- and closed-ended song learning. We mined the literature for longitudinal field and laboratory observations to classify species as exhibiting 'adult song stability' or 'adult song plasticity'. This classification is a quantifiable proxy for closed- and open-ended learning, as the true length of the song-learning window is difficult to assess directly in nature; ultimately, we obtained data for the classification of

67 species. For these species, we also compiled a database of seven species-level song characteristics that can represent either song complexity (syllable repertoire, syllables per song, and song repertoire) or song performance (song duration, inter-song interval, song rate, and song continuity). We then performed phylogenetically controlled analyses to evaluate the evolution of song and mating strategies alongside the relative plasticity of song over time. We find that adult song plasticity has evolved numerous times in bird species. Further, we find evidence of correlated evolution between adult song stability and plasticity and social mating system, with shifts in social mating system occurring more rapidly in lineages with adult song plasticity. In addition, we find a significant evolutionary pattern: species with plastic songs generally have larger repertoires than species with stable songs. Specifically, the evolution of larger syllable and song repertoires appears to drive an evolutionary transition toward open-ended learning.

## Results

In this study, we analyzed the length of the song-learning window using three classification schemes. First, we classified 67 species as having either adult song plasticity (i.e. those that change their syllable repertoire after their first breeding season ends) or adult song stability. Second, when possible, we reclassified the species into three categories: 1) those that stop changing their song before their first breeding season, 2) those that modify their songs during their first breeding season but not after, and 3) those that modify their songs after their first breeding season has ended. This three-state classification was possible for 59 out of the 67 species. Third, for the same 59 species, we used reported estimates of the ages at which song stabilizes to create a continuous measure of the song-learning window. There were exceedingly few studies that examined song changes after the second breeding season, so our continuous metric ranges from 0 to 2 years. We then examined the evolutionary patterns of adult song stability and plasticity, as well as their interactions with species-level song characteristics and mating behaviors.

### How has adult song plasticity evolved across clades?

We were first interested in examining the rate of evolution of adult song stability versus adult song plasticity, as well as when and where evolutionary transitions in these traits occurred on a phylogenetic tree (*Jetz et al., 2012*) using ancestral state reconstruction. As with any reconstruction of evolutionary history, these simulations cannot exactly predict the ancestral states but aim to approximate them. Furthermore, we note that only a subset of oscine families were represented in our analysis, and most of the early branching lineages that would be required to assess the ancestral state for all oscine species were not included in our dataset. Ultimately, we could not make a conclusion about whether the last common ancestor for the species included in this study had adult song plasticity, but our results hint that there might have been several early transitions in this trait, leading to clades that predominantly have adult song stability or plasticity, coupled with a number of more recent transitions (see pie charts in *Figure 1A* for the predicted likelihood of each state at each node). We found that a model allowing the transition rate from song stability to plasticity to be different from the transition rate from plasticity to stability (all-rates-different model [ARD]) did not fit the data significantly better than a simpler model allowing for only one rate of transition back and forth between song stability and plasticity (equal rates model [ER]) (LogLikelihood$_{ER}$ = $-38.22$, LogLikelihood$_{ARD}$ = $-38.21$, p=0.87). At least 14 transitions were required to explain the current binary song-stability states of our subset of bird species. Explaining the distribution of song plasticity in our subset of species most parsimoniously requires at least nine transitions to adult song plasticity if the last common ancestor was song-stable and seven transitions to song stability if the common ancestor was song-plastic (*Figure 1—figure supplement 1*).

### Which song traits differ between species with stable versus plastic songs?

We next tested whether song characteristics were affected by the length of the song-learning window on an evolutionary scale. Intuitively, it makes sense that a species that has a longer time-window to learn might be able to accumulate a larger repertoire. Indeed, this relationship is consistent with the pattern of song stability and repertoire size in several clades, such as the *Phylloscopus* species (*Figure 2*). However, many individual species do not follow this prediction: for example,

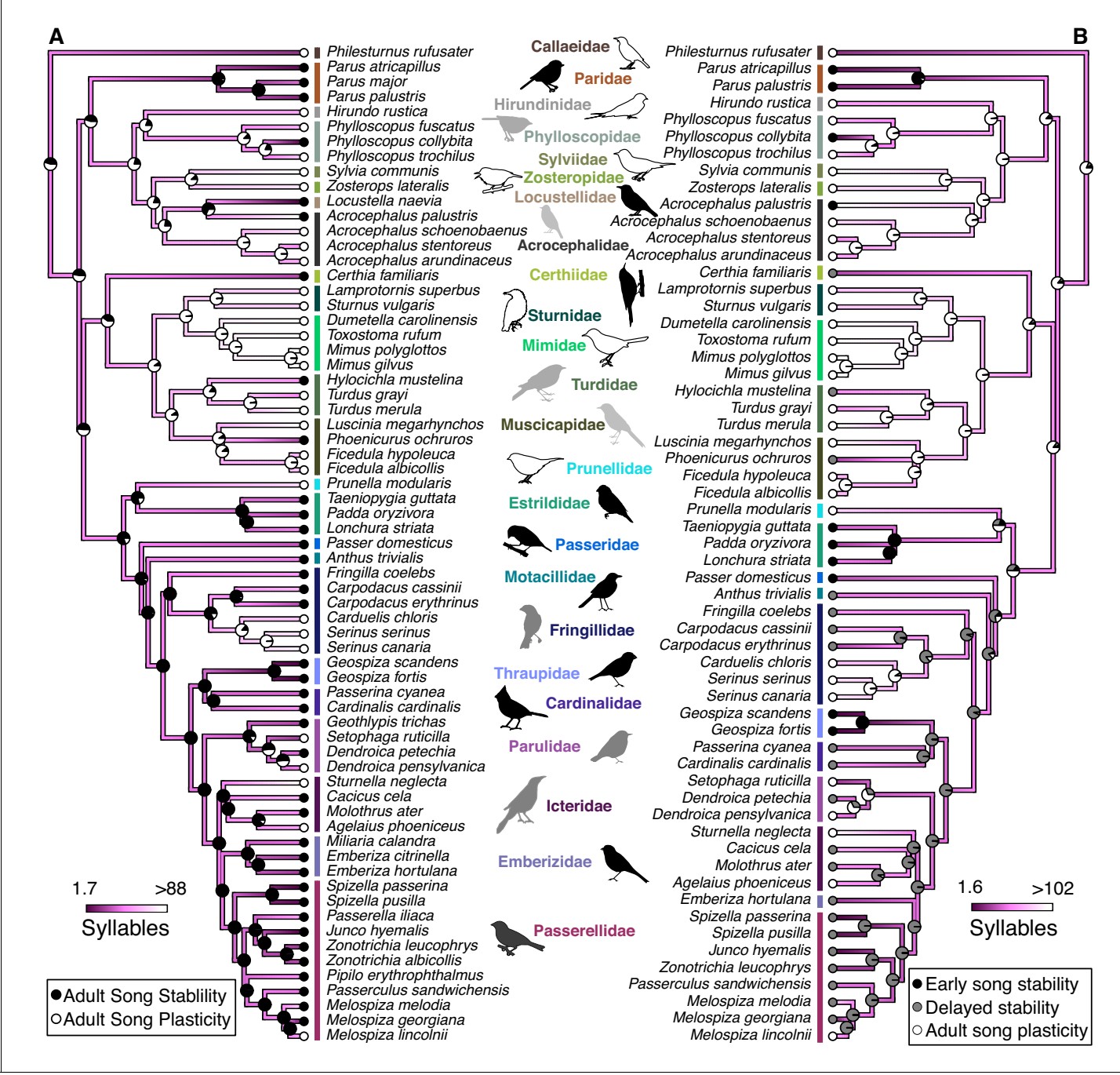

**Figure 1.** Syllable repertoire size is larger in species with adult song plasticity even when controlling for phylogenetic relationships. These phylogenies show the calculated evolution of natural-log transformed syllable repertoire size and either (**A**) stable and plastic song stability states or (**B**) early song-stable, delayed song-stable, and song-plastic states. Dots at the tips of branches represent the current song-stability state. Pie charts represent the likelihood that the common ancestor at that node was in each song-stability state. Dark purple colors represent small syllable repertoires while white represents large repertoires. For the sake of visualization, the color range was truncated based on the distribution of the data, such that the lowest value was the 25th percentile minus the range of the 25th to 50th percentile and the highest value was the 75th percentile plus the range of the 50th to 75th percentile. See *Table 1* for PhylANOVA results.

DOI: https://doi.org/10.7554/eLife.44454.003

The following source data and figure supplements are available for figure 1:

**Source data 1.** Predicted values for internal nodes for each trait.

DOI: https://doi.org/10.7554/eLife.44454.012

*Figure 1 continued on next page*

*Figure 1 continued*

**Figure supplement 1.** Minimum number of evolutionary transitions required to recapitulate the current song stability states of birds in this study.
DOI: https://doi.org/10.7554/eLife.44454.004

**Figure supplement 2.** There was a significant relationship between song repertoire size and song stability when controlling for phylogeny.
DOI: https://doi.org/10.7554/eLife.44454.005

**Figure supplement 3.** There was no relationship between syllables per song and song stability when controlling for phylogeny.
DOI: https://doi.org/10.7554/eLife.44454.006

**Figure supplement 4.** There was no relationship between song duration and song stability when controlling for phylogeny.
DOI: https://doi.org/10.7554/eLife.44454.007

**Figure supplement 5.** There was no relationship between intersong interval and song stability when controlling for phylogeny.
DOI: https://doi.org/10.7554/eLife.44454.008

**Figure supplement 6.** There was no relationship between song rate and song stability when controlling for phylogeny.
DOI: https://doi.org/10.7554/eLife.44454.009

**Figure supplement 7.** There was no relationship between song continuity and song stability when controlling for phylogeny.
DOI: https://doi.org/10.7554/eLife.44454.010

**Figure supplement 8.** There was no significant correlation between the rate of syllable repertoire evolution and ancestral syllable repertoire size in closely related species pairs (Pearson's $r$ = 0.101, p=0.6715).
DOI: https://doi.org/10.7554/eLife.44454.011

*Acrocephalus palustris* appears to learn a large repertoire in a single year (*Dowsett-Lemaire, 1979*), and *Philesturnus rufusater* modifies its song for multiple years but maintains a small repertoire (*Jenkins, 1978*). Further, numerous species with adult song plasticity do not significantly increase their repertoire sizes over time (*Eriksen et al., 2011*; *Galeotti et al., 2001*; *Garamszegi et al., 2005*; *Nicholson et al., 2007*). Thus, an evolutionary link between adult song plasticity and larger repertoire sizes cannot be assumed. Using a phylogenetically controlled ANOVA (*Garland et al., 1993*; *Revell, 2012*), we found that species with adult song plasticity did possess significantly larger syllable repertoires than species with adult song stability (*Figures 1A* and *3A*, *Table 1*). This concurs with a previous analysis using a smaller dataset (*Creanza et al., 2016*). Similarly, we found that song-plastic species had significantly larger song repertoires than song-stable species (*Figure 3B*, *Figure 1—figure supplement 2*, and *Table 1*).

There were no significant differences between song-plastic and song-stable species for the other song characteristics that we tested: syllables per song, inter-song interval, song duration, song rate (calculated as 60/(interval + duration)), or song continuity (calculated as duration/(duration + interval)) (*Table 1*, *Figure 1—figure supplements 3–7*). When we used the classification scheme with three states, we could only test for differences in syllable repertoire, song repertoire, and syllables per song between groups, as there were very few early song-stable species for which we had data on the other song traits. We found no significant differences between early song-stable and delayed song-stable species for any tested traits, but both of these groups had significantly smaller syllable and song repertoires compared to song-plastic species (*Figures 1B* and *3C*, *Tables 2* and *3*). When performing a phylogenetic generalized least squares (PGLS) analysis using continuous estimates of the duration of song plasticity, we found similar results; both syllable repertoire and song repertoire were correlated with duration of song plasticity, such that repertoire size increased with the song-plasticity duration (*Figure 3D*, *Table 4*).

## Do song characteristics evolve at different rates in song-stable or song-plastic species?

Our result that species with adult song plasticity had significantly larger syllable and song repertoires raised the question of whether song stability versus plasticity also affected the rate of evolution for any of the song characteristics. To examine this possibility, we used the Brownie algorithm (*O'Meara et al., 2006*), which tests whether a model with two rates of evolution for each song characteristic—one rate for ancestral periods of song stability and another rate for song plasticity—fits the data significantly better than a model that allows for only a single rate of evolution of each song characteristic regardless of the ancestral states of song stability. Each calculation of the two-rate model is based on one stochastic projection of the ancestral traits across the phylogenetic tree, so we generated 1300 different stochastic simulation maps to use with Brownie. We plotted the

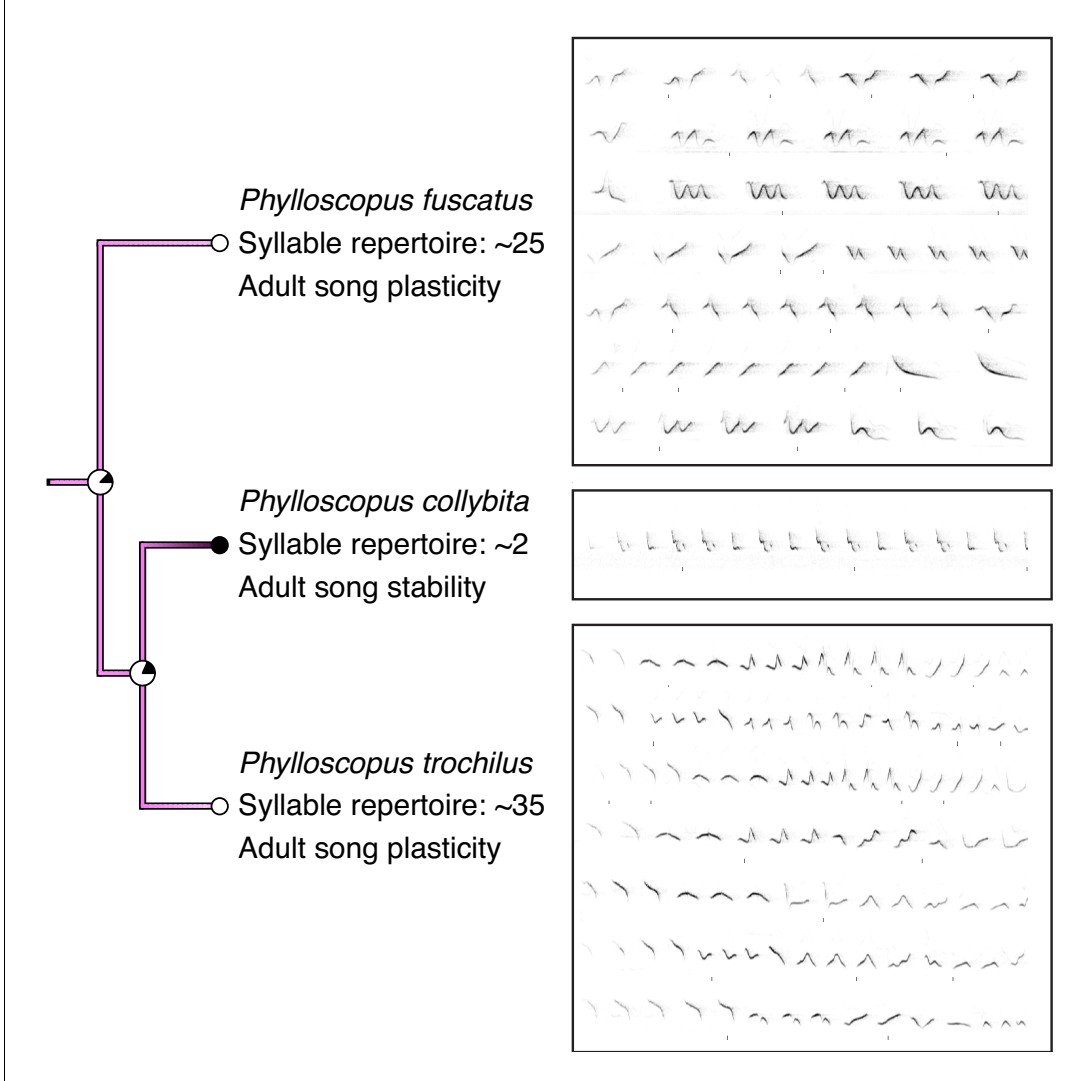

**Figure 2.** Comparison of song stability state and song examples in *Phylloscopus* species. *P. collybita*, a species with adult song stability, has a smaller syllable repertoire size than *P. trochilus* and *P. fuscastus*, two species with adult song plasticity. Colors of branches and nodes correspond with *Figure 1*. Sonograms were generated from recordings obtained from xeno-canto.org: XC340281 recorded by Tom Wulf (*P. fuscatus*), XC414221 recorded by Frank Lambert (*P. collybita*), and XC402265 recorded by Hans Matheve (*P. trochilus*). Sonograms are used only to demonstrate comparative repertoire size from one individual for each species and were stretched horizontally to fit the allotted space.
DOI: https://doi.org/10.7554/eLife.44454.014

distribution of potential rates (*Figures 4–5*) and compared the average log likelihood of the two-rate models to the log likelihood of the one-rate model (*Table 5*).

We found that allowing for two different rates of song trait evolution depending on song stability or plasticity did not lead to a significantly better fit model than using one Brownian-motion rate for either syllable repertoire size or song repertoire size, even though syllable repertoires and song repertoires were both significantly larger in species with adult song plasticity (*Figure 4A and B* and *Table 5*). In contrast, the two-rate model led to a significantly better fit for syllables per song, song rate, inter-song interval, and song duration (*Figures 4C* and *5* and *Table 5*), indicating that evolution of these song characteristics was faster in song-plastic lineages (*Figures 4C* and *5*, red traces).

We repeated this analysis with the three-state categorization for syllable repertoire, song repertoire, and syllables per song; for other song characteristics, we did not have enough species in the early song stability group. We found that the three-rate model was significantly better than the one-rate model for syllables per song and song repertoire, but not for syllable repertoire (*Figure 4D–F*

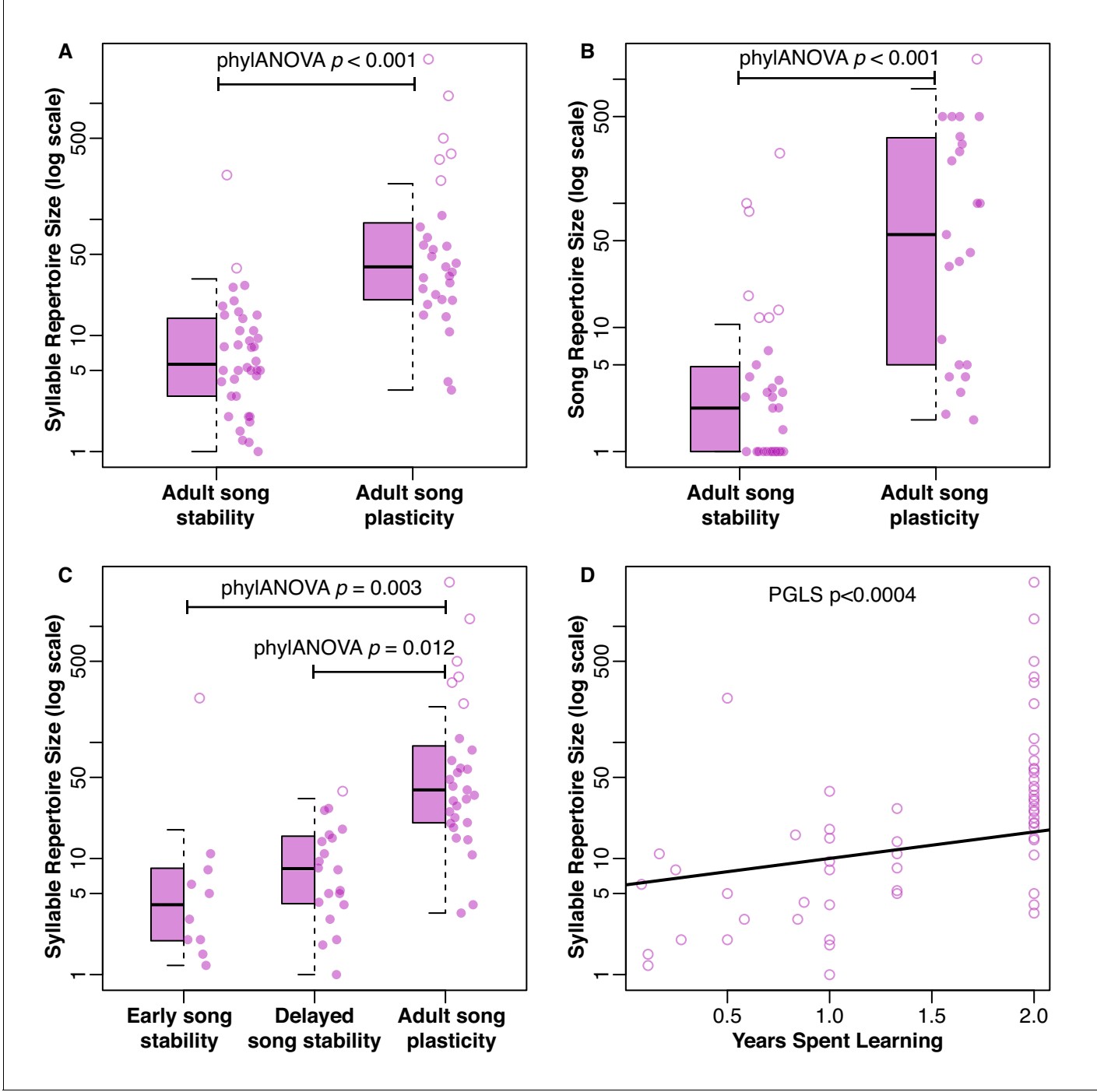

**Figure 3.** Distribution of repertoire sizes in species with different song stability states. (**A**) shows the distribution of syllable repertoires and (**B**) shows the distribution of song repertoires when species are broken into two groups based on song stability. (**C**) shows the distribution of syllable repertoires when species are broken into three groups based on song stability. Boxes indicate the 25th, 50th, and 75th percentile. The lower whisker is either the minimum value or the 25th percentile minus 1.5 times the interquartile range, whichever was larger. The upper whisker is either the maximum value or the 75th percentile plus 1.5 times the interquartile range, whichever was smaller. Dots are the raw values as a scatter plot. Solid dots are within the range of the box and whiskers, while open dots are outliers. (**A-B**) Species with adult song plasticity had significantly larger syllable and song repertoires. See *Table 1* for full PhylANOVA results. (**C**) Species with adult song plasticity had significantly larger syllable repertoires than early song stable and delayed song stable species, but there was no significant difference between early and delayed song stable species (p=0.659). See *Tables 2* and *3* for full PhylANOVA results. (**D**) shows the continuous relationship between syllable repertoire size and years spent learning, where song plasticity is truncated at 2 years due to lack of data from subsequent years.

DOI: https://doi.org/10.7554/eLife.44454.015

**Table 1.** PhylANOVA results for all song traits when birds are divided into species with adult song stability or adult song plasticity.

Song traits are sorted from most to least significant. Song-stable and song-plastic columns show mean values of each log-transformed song trait. Corrected α indicates the threshold for significance with the Holm-Bonferroni correction.

| Song trait | Song-stable | Song-plastic | F-Value | Corrected α | p-value |
|---|---|---|---|---|---|
| Syllable repertoire | 1.8807 | 3.946 | 41.5064 | 0.0071 | <0.001* |
| Song repertoire | 1.1055 | 3.8688 | 33.8334 | 0.0083 | <0.001* |
| Syllables/song | 1.2556 | 2.2962 | 9.2658 | 0.01 | 0.094 |
| Duration | 0.7736 | 1.2927 | 2.0783 | 0.0125 | 0.42 |
| Continuity | -1.3453 | -1.0286 | 2.1537 | 0.0167 | 0.474 |
| Interval | 1.6075 | 1.218 | 1.3879 | 0.025 | 0.567 |
| Song rate | 1.8969 | 2.0971 | 0.6079 | 0.05 | 0.713 |

*Denotes traits with significantly different groups.

DOI: https://doi.org/10.7554/eLife.44454.013

and *Table 6*). However, the three-rate model was only significantly better than the two-rate model for song repertoire (*Table 7*). Thus, the two-rate model sufficiently approximated the evolution of syllables per song. We noticed that for both song repertoire and syllable repertoire, the rate of evolution in delayed song-stable lineages (purple traces in *Figure 4D,E*) was very similar to the rate in song-plastic lineages (corresponding red traces). We tested one more set of models where we combined delayed song-stable species with song-plastic species to create a 'longer learning' group, while early song-stable species were assigned to a 'shorter learning' group. For this comparison of shorter versus longer learning, the two-rate model was significantly better than the one-rate model for song repertoire and trending in that direction for syllable repertoire (*Figure 4G,H* and *Table 8*). The three-rate model was not significantly better than the longer/shorter-learning two-rate model for either syllable or song repertoire (*Table 9*). Taken together with our phylANOVA results, this pattern suggests that species with early song stability evolve their song repertoires and potentially their syllable repertoires at a slower rate than delayed song-stable and song-plastic species; however, only song-plastic species directionally evolve towards larger song and syllable repertoires.

## Is song stability or plasticity influenced by the evolution of song traits and mating strategies, or vice versa?

While the Brownie algorithm tested whether adult song plasticity affected the rate of evolution for song characteristics, it did not address whether the opposite might be true. We used BayesTraits (*Pagel, 1994*; *Pagel and Meade, 2006*) to test whether the rate and order of evolutionary transitions in one trait is dependent on the state of another trait. Because the song features were continuous variables, we binarized them by setting a series of threshold values to delineate 'low' and 'high' categories, using each observed song feature value as a threshold in turn. We then tested whether

**Table 2.** PhylANOVA results for all song traits when birds are divided into early song stability, delayed song stability, and song plasticity.

Song traits are sorted from most to least significant. Early, delayed, and plastic columns show mean values of each log-transformed song trait. Corrected α indicates the threshold for significance with the Holm-Bonferroni correction.

| Song trait | Early | Delayed | Plastic | F-Value | Corrected α | p-value |
|---|---|---|---|---|---|---|
| Syllable repertoire | 1.6436 | 2.0062 | 3.946 | 17.1099 | 0.0071 | 0.003* |
| Song repertoire | 0.6788 | 1.4819 | 3.8688 | 12.88 | 0.0083 | 0.011* |
| Syllables/song | 1.2852 | 1.2467 | 2.2962 | 3.6877 | 0.01 | 0.252 |

*Denotes traits with significantly different groups.

DOI: https://doi.org/10.7554/eLife.44454.016

**Table 3.** Post-hoc pairwise phylANOVA tests for significant song traits when birds are divided into early song stability, delayed song stability, and song plasticity.

| Song trait | State 1 | State 2 | T-Value | p-value |
|---|---|---|---|---|
| Syllable repertoire | Plastic | Delayed | 4.8995 | 0.012* |
| Syllable repertoire | Early | Plastic | 4.6091 | 0.003* |
| Syllable repertoire | Early | Delayed | 0.6872 | 0.659 |
| Song repertoire | Plastic | Delayed | 4.0268 | 0.044* |
| Song repertoire | Early | Plastic | 4.3074 | 0.015* |
| Song repertoire | Early | Delayed | 1.0444 | 0.55 |

*Denotes traits with significantly different groups.
DOI: https://doi.org/10.7554/eLife.44454.017

there was correlated evolution between the binary classifications of adult song plasticity versus stability and each of the seven song characteristics.

In the lowest third of syllable repertoire thresholds, adult song plasticity with small syllable repertoires was an evolutionarily unstable state, with rapid transitions primarily toward a song-stable state and secondarily toward larger repertoires (82% of runs significant in this range, *Figure 6*). In the middle third of syllable repertoire thresholds, song stability with smaller syllable repertoires is an evolutionarily stable attractor state, with high rates of transition observed from large to small syllable repertoires in song-stable species and from plasticity to stability with a small syllable repertoire. These rate differences are highly significant (100% of runs significant in this range). In the highest third of syllable repertoire thresholds, adult song stability with a large syllable repertoire is an evolutionarily unstable state, transitioning primarily toward adult song plasticity (86% of runs significant in this range, *Figure 6*). We found similar trends when using two, four, and five bins for the song characteristic threshold values, with subtle differences. When using four or five bins, we still observe that song stability with larger syllable repertoires is an unstable combination. However, for the highest bin of threshold values, the transition rates are faster when changing to song plasticity, whereas for the second-highest bin, we observe faster transition rates toward repertoire size increases (*Figure 6—source data 1*). To rule out the possibility that syllable repertoire size evolution is faster in species with larger repertoire sizes regardless of learning program, we tested the rates of evolution of syllable repertoire size across monophyletic species pairs in our dataset. We found that lineages with larger syllable repertoire sizes do not systematically undergo faster or slower syllable repertoire size evolution (*Figure 1—figure supplement 8*).

At low song repertoire threshold values, song plasticity with a small repertoire is an evolutionarily unstable state; there is rapid transition away from this combination, predominantly trending toward

**Table 4.** Results of PGLS analysis between song characteristics and continuous song stability.
Test performed on the natural-log scaled values of song characteristics. λ is the value by which off-diagonal elements in the Brownian motion model are multiplied to make the correlation structure. Corrected α indicates the threshold for significance with the Holm-Bonferroni correction. Song traits are sorted from most to least significant.

| Song trait | Slope | Std error | λ | T-Value | Corrected α | p-value |
|---|---|---|---|---|---|---|
| Syllable repertoire | 0.9067 | 0.2449 | 0.8913 | 3.7021 | 0.0071 | <0.001* |
| Song repertoire | 1.1013 | 0.3123 | 0.8316 | 3.5263 | 0.0083 | <0.001* |
| Syllables/song | 0.3701 | 0.2224 | 0.4699 | 1.6642 | 0.01 | 0.1029 |
| Interval | 0.4221 | 0.2646 | 0.8823 | 1.5953 | 0.0125 | 0.1215 |
| Continuity | -0.2135 | 0.1439 | 0.8832 | -1.4838 | 0.0167 | 0.1486 |
| Duration | 0.3702 | 0.2569 | 1.0163 | 1.441 | 0.025 | 0.1578 |
| Song rate | -0.2113 | 0.25 | 0.7307 | -0.8453 | 0.05 | 0.4048 |

*Denotes significant slopes.
DOI: https://doi.org/10.7554/eLife.44454.018

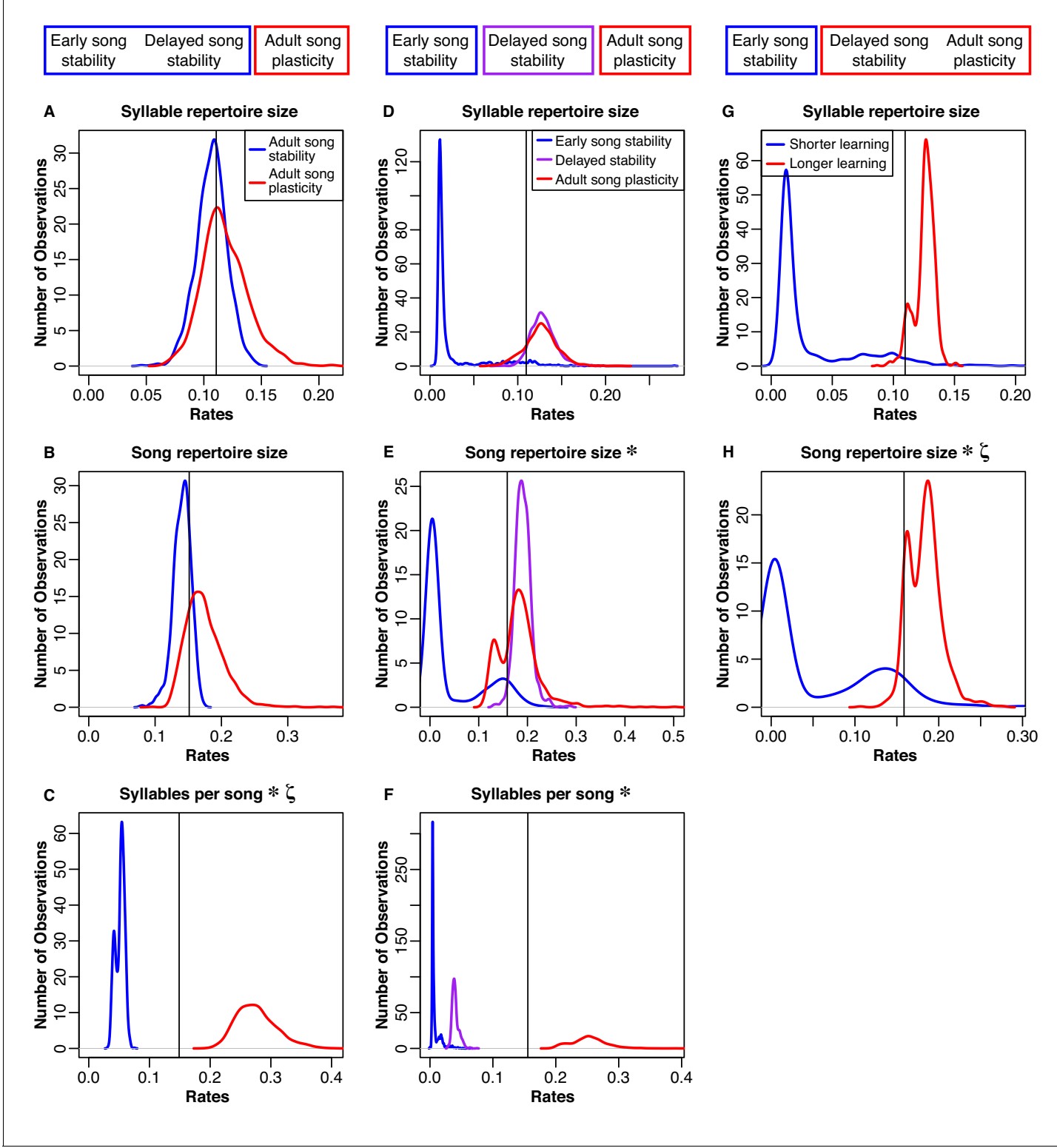

**Figure 4.** Distributions of rates for natural-log transformed song traits related to complexity. The boxes at the top illustrate how we grouped the species for each model. Column 1 (**A-C**): Blue traces are song-stable, while red traces are song-plastic. Column 2 (**D-F**): Blue traces are early song-stable, purple traces are delayed song stable, and red traces are song-plastic. Column 3 (**G-H**): Blue traces are early song-stable, while red traces are delayed song-stable and song-plastic combined. The black line shows the rate value for the one-rate model in all columns. Asterisks indicate that the rate of evolution of that song characteristic significantly differed between groups. Lowercase zeta (ζ) the multi-rate model that best fit the data while

*Figure 4 continued on next page*

*Figure 4 continued*

using the fewest number of rates. In the case of syllable repertoire, the multi-rate models were not significantly better than the one-rate model. See *Tables 5–9* for chi-square test results.

DOI: https://doi.org/10.7554/eLife.44454.020

song stability but also transitioning secondarily to a larger song repertoire, with very high significance (100% of runs significant in this range). At moderate song repertoire thresholds, the highest rate is observed for species with small repertoires transitioning from song plasticity to song stability, also with very high significance (100% of runs significant in this range). At high song repertoire thresholds, the primary shift is from song stability to song plasticity in species with large song repertoires (89% of runs significant in this range) (*Figure 7*). When analyzing the results using five bins, transitions in the upper range of song repertoire values becomes more nuanced; in the highest bin, song stability with a larger song repertoire is very unstable, but is relatively stable in the second-highest bin. In the lower four bins, the dominant transition is from plastic to stable song with smaller song repertoires (*Figure 6—source data 1*). The results for syllables per song show some general trends that are complicated by the strong effect of the mimid species (see results of jackknife analysis below). We did not find evidence for correlated evolution between adult song stability or plasticity and any of the other song characteristics (*Figure 6—figure supplements 1–5*).

In addition, it has been proposed that polygyny and extra-pair paternity (EPP) may increase sexual selection pressures on sexually selected traits, including song (*Emlen and Oring, 1977*; *Payne, 1984*), and increased selection pressure due to polygyny was theorized to accelerate the evolution of song learning in a mathematical model (*Aoki, 1989*). We tested for correlated evolution between adult song plasticity versus stability and both social mating system (polygyny vs. monogamy) and extra-pair paternity (low vs. high EPP), with the caveat that many species in our dataset lacked mating behavior classifications (57 species with social mating system data, 41 with EPP data). We did not find evidence for correlated evolution between song stability and EPP. There was, however, evidence for correlated evolution between polygynous/monogamous mating systems and song plasticity (100% of runs significant), with elevated rates of transition between polygyny and monogamy in the song-plastic state (*Figure 6—figure supplement 6*).

## Do our results depend on the particular values or families included in analyses?

In many cases, there were multiple studies that gave different estimates for a given song trait in one species, so we used the median values across studies for our main analysis. To test whether our results depended on the particular values we used, we repeated the PhylANOVA and Brownie analysis using the either the maximum or minimum values reported in the literature. This did not significantly alter our PhylANOVA results for any song feature when species were split into song-stable and song-plastic groups (*Supplementary file 1* Table S1), though when the early song-stable, delayed song-stable, and song-plastic dataset was used, using the maximum values for song repertoire led to non-significant results (*Supplementary file 1* Table S2-3). When species were split into those with adult song plasticity and adult song stability, the Brownie analysis suggested that syllables per song was evolving significantly faster in song-plastic lineages when we used the median value, but using the minimum values led to non-significant results (*Supplementary file 1* Table S4). When species were split into three states (early song stability, delayed song stability, and song plasticity), we found that the three-rate model was not better than the one-rate model when the minimum values for song repertoire were used, while the three-rate models for syllable repertoire became significant when either the maximum or the minimum values were used (*Supplementary file 1* Table S5).

In our dataset, 24 songbird families were represented by 1 to 11 species each. This meant that families had unequal influence on the outcomes of the analyses. We performed a jackknife analysis to examine whether our results were affected by excluding individual families represented by four or more species in the full dataset. Exclusion of individual families did not significantly alter any of the phylANOVA results (*Supplementary file 1* Tables S6-11). In general, exclusion of individual families did not affect the Brownie results (*Supplementary file 1* Tables S12-17) or the BayesTraits results

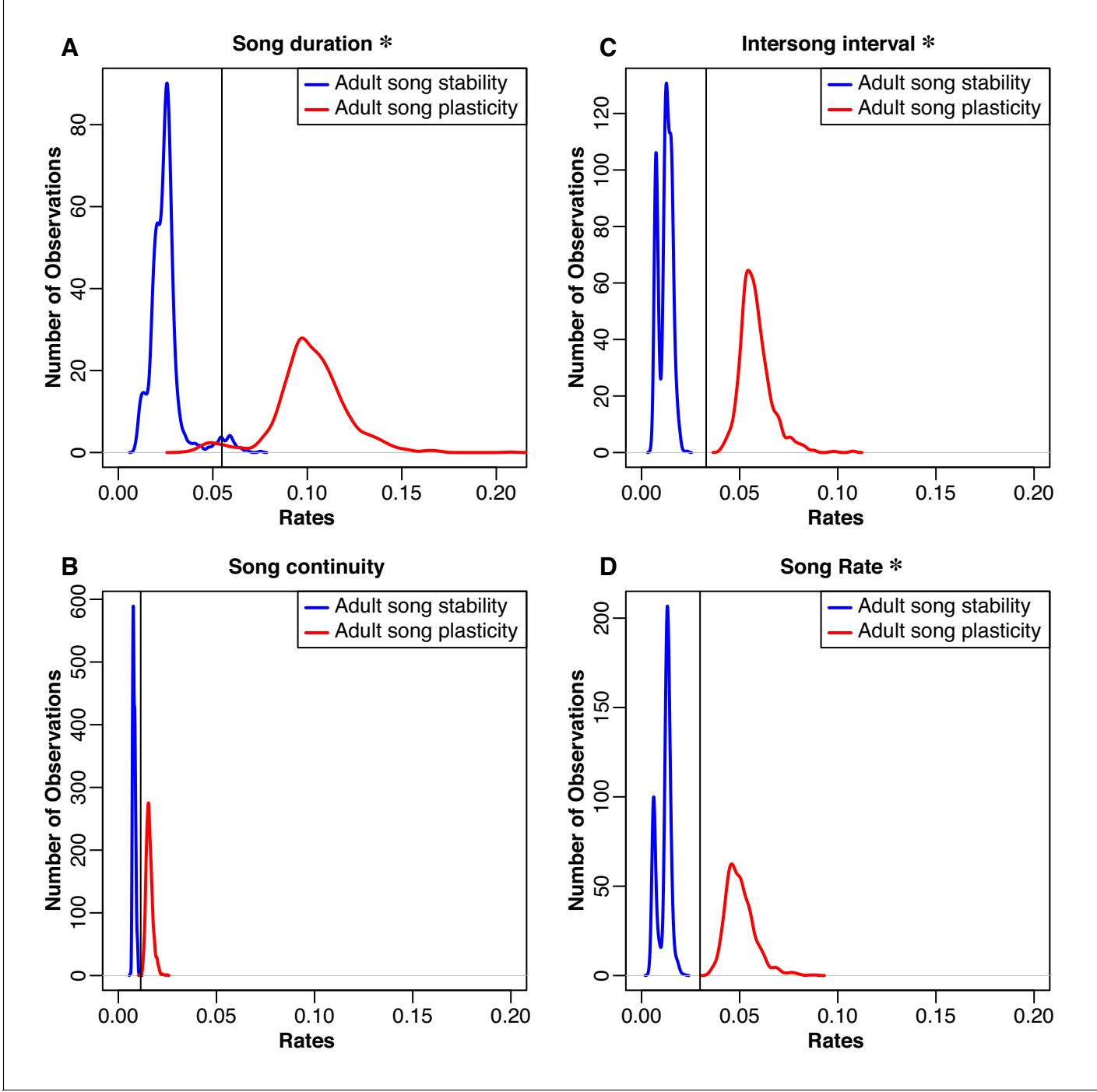

**Figure 5.** Distributions of rates of evolution for natural-log transformed song traits related to performance. Blue traces song-stable, while red traces are song-plastic. The black line shows the rate for the one-rate model. Asterisks indicate that the rate of evolution of that song characteristic significantly differed between song-stable and song-plastic lineages. See *Table 5* for chi-square test results.

DOI: https://doi.org/10.7554/eLife.44454.021

(*Figure 6—source datas 2–3*, *Figure 7—source data 1*). However, there were several notable exceptions, detailed below.

In several cases, removing a single family altered the significance of our findings. Removal of the Muscicapidae (three species) from the Brownie analysis of inter-song interval created a two-rate model that did not fit the data significantly better than the one-rate model (*Supplementary file 1*

**Table 5.** Brownie results for song traits when birds are divided into species with adult song stability or adult song plasticity.

Rate columns show mean log likelihood. Song traits are sorted from most to least significant.

| Song trait | One rate | Two rates | p-value |
|---|---|---|---|
| Syllables/song | -110.6482 | -100.7673 | <0.001* |
| Song rate | -43.4397 | -38.4938 | 0.002* |
| Interval | -45.2842 | -40.5004 | 0.002* |
| Duration | -71.2042 | -66.3122 | 0.002* |
| Continuity | -25.6471 | -24.7285 | 0.175 |
| Syllable repertoire | -120.2983 | -120.0695 | 0.499 |
| Song repertoire | -113.5829 | -113.3706 | 0.515 |

*Denotes traits where the more complex model fit the data significantly better than the simpler model.
DOI: https://doi.org/10.7554/eLife.44454.019

Table S13). For song duration, removal of Fringillidae (three species) led to a two-rate model that was not a significantly better fit than the one-rate model (*Supplementary file 1* Table S14). Removal of Mimidae (four species) from the Brownie analysis of syllables per song drastically changed the results, such that the two-rate model was no longer a significantly better fit than the one-rate model (*Supplementary file 1* Table S15). All of the included Mimidae species are song-plastic, so these results suggest that mimids may be driving the the fit of the two-rate model for syllables per song. In addition, removal of Mimidae from the BayesTraits analysis of song plasticity and syllables per song altered the observed trends. At low threshold values of syllables per song, song plasticity with low syllables per song is an unstable state, with high rates of transition toward either higher syllables per song or toward song stability (77% of runs significant in this range); however, when mimid species are removed, gaining more syllables per song in the song-plastic state becomes far more likely. At moderate values of syllables per song, there are elevated rates of transition both toward more syllables per song and towards fewer syllables per song in song-plastic species (100% of runs significant in this range); there is a comparable trend when mimid species are removed. At high threshold values, more syllables per song with song stability is an unstable state (63% of runs significant in this range, *Figure 6—figure supplement 2*); however, with mimids removed, more syllables per song with song plasticity becomes the most unstable state (*Figure 6—source data 3*).

Due to the strong effect that the inclusion of Mimidae had on the Brownie analysis of syllables per song, we performed a second jackknife analysis, in which each of the four mimid species was removed in turn. Exclusion of *Toxostoma rufum* or *Dumetella carolinensis* had little effect on the results (*Supplementary file 1* Table S18), leading to significant support for the two-rate model. However, exclusion of *Mimus polyglottos* (p=0.074) or *Mimus gilvus* (p=0.509) led to a two-rate model that did not fit the data significantly better than the one-rate model (*Supplementary file 1* Table S18). Therefore, we concluded that these two *Mimus* species drove the estimated evolutionary rate of syllables per song in song-plastic species to be much greater than in song-stable species,

**Table 6.** Brownie results for song traits when birds are divided into early song stability, delayed song stability, and song plasticity.

Rate columns show mean log likelihood. Song traits are sorted from most to least significant.

| Song trait | One rate | Three rates | p-value |
|---|---|---|---|
| Syllables/song | -97.8349 | -86.3206 | <0.001* |
| Song repertoire | -100.812 | -97.7647 | 0.014* |
| Syllable repertoire | -107.3206 | -105.5895 | 0.063 |

*Denotes traits where the more complex model fit the data significantly better than the simpler model.
DOI: https://doi.org/10.7554/eLife.44454.022

**Table 7.** Brownie results for song traits when birds are divided into either song stability (early plus delayed) and song plasticity (Two Rates) or early song stability, delayed song stability, and song plasticity (Three Rates).

Rate columns show mean log likelihood. Song traits are sorted from most to least significant.

| Song trait | Two rates | Three rates | *p*-value |
| --- | --- | --- | --- |
| Song repertoire | -100.691 | -97.7148 | 0.015* |
| Syllable repertoire | -107.1332 | -105.5532 | 0.075 |
| Syllables/song | -86.3125 | -86.3447 | 1 |

*Denotes traits where the more complex model fit the data significantly better than the simpler model.
DOI: https://doi.org/10.7554/eLife.44454.023

and that faster evolution of syllables per song may not necessarily be a universal trend for song-plastic species.

Members of Mimidae are renowned for their vocal mimicry, frequently exhibiting improvisation and invention of syllables beyond simple imitation, and thus they lack the generally stereotyped song structure shown in other oscine families. Furthermore, mimids often have periods of continuous singing with minimal repetition of elements and irregular syllable spacing. Thus, quantification of song duration or number of syllables per song for mimids could be highly susceptible to listener perception (*Wildenthal, 1965*). Therefore, although we acknowledge that mimids are an important case study in extended learning durations, our results for the evolutionary rate of syllables per song might be more meaningful across all bird species when mimids are excluded, in which case we find the rate of evolution of syllables per song is be independent of adult song plasticity.

## Discussion

Previously, it was unknown whether the song-learning window evolved in concert with song features associated with sexual selection, as predicted by a computational model of song learning (*Creanza et al., 2016*). This is a critical missing piece of the puzzle of song learning evolution, as previous evidence has suggested that sexual selection only acts upon the features of song and not the length of the song-learning window or maintenance of the song-learning pathways in the brain. Here, we performed phylogenetically controlled analyses to assess the interactions between the length of the song-learning window—using adult song stability versus plasticity as a proxy—and the evolution of song characteristics. Interestingly, we noted that several evolutionary events relatively early in passerine evolution accounted for much of the diversity in the song-plasticity period in our sample of species. We show that a bird's ability to modify its song as an adult affects the characteristics of its species' song: adult song plasticity corresponds to larger syllable and song repertoires. Further, our results suggest that sexual selection for large repertoires could indirectly favor individuals with longer learning windows, driving the evolution of increased song plasticity.

We found two key trends in the trait correlation (phylANOVA) and evolutionary rate (Brownie) analyses. First, song plasticity affected the *direction* of evolution in traits that can be considered metrics of song complexity (syllable and song repertoire size, *Figures 1* and *3*, *Table 1*), leading to larger repertoires in species with adult song plasticity. Further, species with early song stability

**Table 8.** Brownie results for song traits when birds are divided into shorter learning (early song stability) and longer learning (delayed song stability plus song plasticity).

Rate columns show mean log likelihood. Song traits are sorted from most to least significant.

| Song trait | One rate | Two rates | *p*-value |
| --- | --- | --- | --- |
| Song repertoire | -100.812 | -97.9918 | 0.018* |
| Syllable repertoire | -107.3206 | -105.8488 | 0.086 |

*Denotes traits where the more complex model fit the data significantly better than the simpler model.
DOI: https://doi.org/10.7554/eLife.44454.024

**Table 9.** Brownie results for song traits when birds are divided into either shorter learning (early song stability) and longer learning (delayed song stability plus song plasticity) (Two Rates) or early song stability, delayed song stability, and song plasticity (Three Rates).
Rate columns show mean log likelihood. Song traits are sorted from most to least significant.

| Song trait | Two rates | Three rates | *p*-value |
|---|---|---|---|
| Syllable repertoire | -105.8156 | -105.5532 | 0.469 |
| Song repertoire | -97.9372 | -97.7148 | 0.505 |

*Denotes traits where the more complex model fit the data significantly better than the simpler model.
DOI: https://doi.org/10.7554/eLife.44454.025

evolved their repertoires at a slower rate than species with longer learning (delayed song stability and song plasticity, *Table 5*), but song-plastic species did not evolve their repertoires at a faster rate than species with early or delayed song stability combined (*Tables 6–9*). Thus, while repertoires only evolve directionally in song-plastic species (*Tables 2–3*), our results suggest that extended learning through the first breeding season allows for faster, but nondirectional, evolution of repertoire size. A possible explanation is that delayed song learning allows individuals to modify their songs after migration and thus adapt their song to their new surroundings once they establish a territory, without necessarily corresponding with sexual selection for larger repertoires. This ability for an individual to adapt to a new local song might be beneficial, particularly when species have local dialect structure; however, this would not lead to directional evolution for any particular song feature, consistent with our findings. Second, song plasticity increased the *rate* of evolution primarily in traits that can be considered metrics of song performance (song duration, intersong interval, and song rate *Figure 5*, *Table 2*). While these performance-related song traits evolved faster in song-plastic lineages, this rapid evolution did not lead to significant differences in those song characteristics compared to song-stable lineages. A possible explanation may be that increases in repertoire size necessitate changes to song structure, but multiple structural aspects of song can be altered to accommodate these changes. Thus, there is no overall pattern of directional evolution in these other song characteristics. Alternatively, bird species may be required to adapt to increasing repertoire sizes while maintaining species-specific constraints imposed by innate aspects of song structure or female preferences for different performance characteristics. In the latter case, if information about innate characteristics and female preferences are known, it may be possible to predict how song traits will change in response to increasing repertoire sizes and greater adult song plasticity.

With our analyses of correlated evolution, we aimed to detect whether the state of the repertoire size or adult song stability versus plasticity consistently changes first in evolutionary history, facilitating a change in the other trait. Overall, our results suggest that there is not a consistent order of evolutionary transitions (*Figures 6–7*). For example, song stability with very large syllable or song repertoires, and song plasticity with very small syllable or song repertoires both formed evolutionarily unstable states, with high evolutionary rates of transition in both repertoire size and song-learning window. However, we do note that the fastest rates of transition in our analyses were those switching between song stability and song plasticity to leave those unstable states. This trend suggests that the magnitude of a species' repertoire may be more likely to drive the evolution of learning window than vice versa (*Figures 6–7*). This is consistent with the idea that selection acting upon song features could indirectly place selective pressures on the learning window. We propose several hypotheses that could explain these evolutionary dynamics: 1) it may be disproportionately costly to maintain song plasticity when syllable or song repertoire sizes are very small, perhaps because the benefit of extra time to learn does not outweigh the metabolic cost of maintaining plasticity, or 2) species with small syllable or song repertoires may have highly stereotyped songs which are selected for based on accuracy of learning and consistency of song production, favoring males that only learn from their fathers or early-life neighbors. Alternatively, in species where females prefer larger repertoires, 3) it may simply require more time to learn a large song or syllable repertoire than is available with a short learning window, or 4) learning large repertoires may require too much energy devoted to song practice during the crucial period of development before and during fledging, favoring birds that can learn for longer periods. Further research into the physiological or reproductive costs of song plasticity is needed.

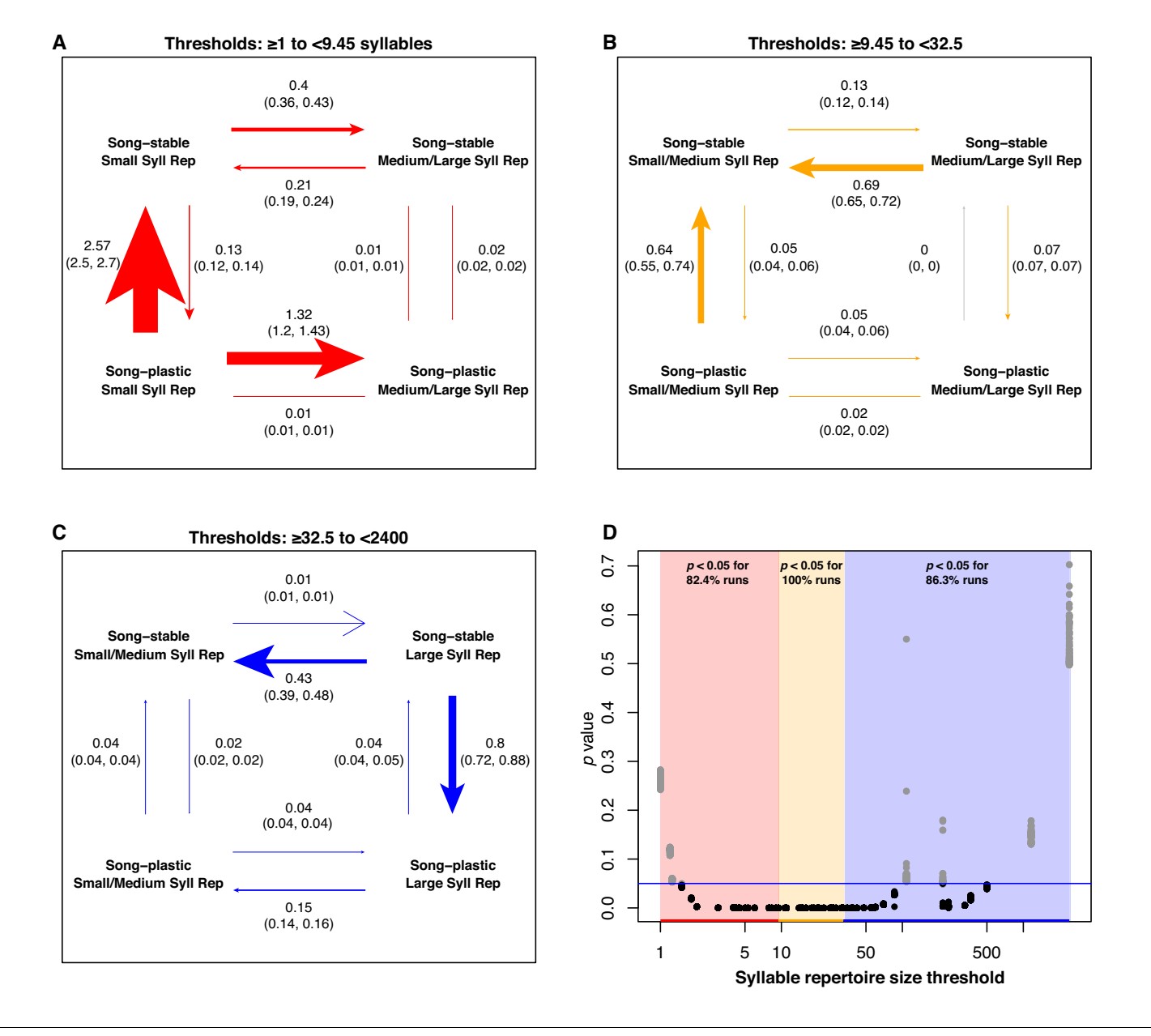

**Figure 6.** Analysis of correlated evolution between adult song stability/plasticity and syllable repertoire size. We repeated the BayesTraits analysis using each value of the continuous song trait as the threshold delineating the larger and smaller syllable repertoire groups. We performed a total of 100 runs per threshold. We pooled the results of all the runs into three groups based on whether the threshold was in the lowest, middle, or highest third of the unique trait values. Within these groups, we computed the mean percentage of runs that were significant at p<0.05 at each threshold. (A-C) Rate of transition plots when the lowest (red), middle (yellow), and highest (blue) thirds of the unique syllable repertoire values in the dataset were used as the threshold. Rates are the average across all runs when the threshold denoting small/large repertoire sizes was defined as each value within each segment. Arrows are labeled with the mean rate and the 95% confidence interval. Arrow weights are scaled to the mean rate values. (D) p-values from the 100 runs per threshold, plotted against threshold. Colored bars denote low, middle, and high threshold segments. Blue line denotes p=0.05.
DOI: https://doi.org/10.7554/eLife.44454.026

The following source data and figure supplements are available for figure 6:

**Source data 1.** Mean transition rates from BayesTraits analysis between song stability and song features averaged over 2, 4, and 5 bins.
DOI: https://doi.org/10.7554/eLife.44454.027

**Source data 2.** BayesTraits analysis of syllable repertoire and learning window, jackknifed across families.
DOI: https://doi.org/10.7554/eLife.44454.034

*Figure 6 continued on next page*

*Figure 6 continued*

**Source data 3.** BayesTraits analysis of syllables per song and learning window, jackknifed across families.
DOI: https://doi.org/10.7554/eLife.44454.035
**Figure supplement 1.** BayesTraits analysis on song stability and syllables per song.
DOI: https://doi.org/10.7554/eLife.44454.028
**Figure supplement 2.** BayesTraits analysis on song stability and song duration.
DOI: https://doi.org/10.7554/eLife.44454.029
**Figure supplement 3.** BayesTraits analysis on song stability and intersong interval.
DOI: https://doi.org/10.7554/eLife.44454.030
**Figure supplement 4.** BayesTraits analysis on song stability and song rate.
DOI: https://doi.org/10.7554/eLife.44454.031
**Figure supplement 5.** BayesTraits analysis on song stability and song continuity.
DOI: https://doi.org/10.7554/eLife.44454.032
**Figure supplement 6.** BayesTraits analysis of song stability and mating behaviors performed over 1000 runs.
DOI: https://doi.org/10.7554/eLife.44454.033

Beginning with *Darwin (1872)*, numerous researchers have proposed that polygynous mating systems could lead to amplified sexual selection (*Emlen and Oring, 1977*; *Payne, 1984*). The elevated rates of transition that we observed between social mating systems in song-plastic lineages suggest an interesting hypothesis for further investigation: perhaps having a plastic song-learning program facilitates evolutionary transitions in mating systems.

Our results provide key evidence that sexual selection upon song characteristics might indirectly act upon the song-learning window. We do not fully understand the mechanisms underlying the maintenance or reopening of the song learning window in adulthood, but genetic, environmental, hormonal, and social factors are likely contributors (*Eales, 1987*; *Eales, 1985*; *Kroodsma and Pickert, 1980*; *Morrison and Nottebohm, 1993*; *Nottebohm, 1969*). For example, when zebra finches were reared in isolation, their sensitive periods were lengthened. These isolated birds maintained both gene expression profiles associated with song learning in the song system and high levels of neuronal addition to the HVC (a key region in the song system of the songbird brain) for longer than birds reared with an adult male tutor (*Kelly et al., 2018*; *Wilbrecht et al., 2006*), linking the neural underpinnings of song learning to the length of the song-learning window (*Brenowitz and Beecher, 2005*; *Nordeen and Nordeen, 1990*; *Nordeen, 1997*; *Nottebohm, 1992*). Furthermore, a positive association between HVC volume and song repertoire size has been demonstrated both intraspecifically (in song sparrows; *Pfaff et al., 2007*) and interspecifically (*Devoogd et al., 1993*). In light of our findings that adult song plasticity correlates with an increase in song repertoire size, there is a logical prediction that extended song learning may be associated with increased HVC volume across species. This is an important avenue for future research.

Although our dataset includes species from 24 different songbird families, many families are not represented due to a lack of data about song stability. It will be important to expand this dataset in future studies. It would also be interesting to explore the evolutionary interactions between adult song plasticity and mimicry of heterospecific sounds, which has been observed in Mimidae and numerous other clades across the songbird lineage (*Goller and Shizuka, 2018*). With our current dataset, we could not adequately explore the effects of mimicry on the evolution of song learning outside of the mimids, but the repeated evolution of mimicry makes it a particularly interesting topic for follow-up studies on the length of the song-learning window. In addition, different song metrics that are tailored to mimicry would be important in studying the evolution of vocal mimics and the dynamics of their unique vocal patterns. Furthermore, there is increasing interest in the importance of female song in species, which is more common than previously thought (*Langmore, 1998*; *Odom et al., 2014*). Our dataset includes numerous species wherein females are known to sing at least occasionally (*Langmore, 1998*), but the length of the song-learning window in females has not been assessed in any of these species. There is, however, some evidence that female birds can modify their song preferences in adulthood (*Nagle and Kreutzer, 1997*). Thus, it remains an open question whether song plasticity affects the evolution of female song in the same way it affects male song, and whether species with adult song plasticity in males also have adult song plasticity in females.

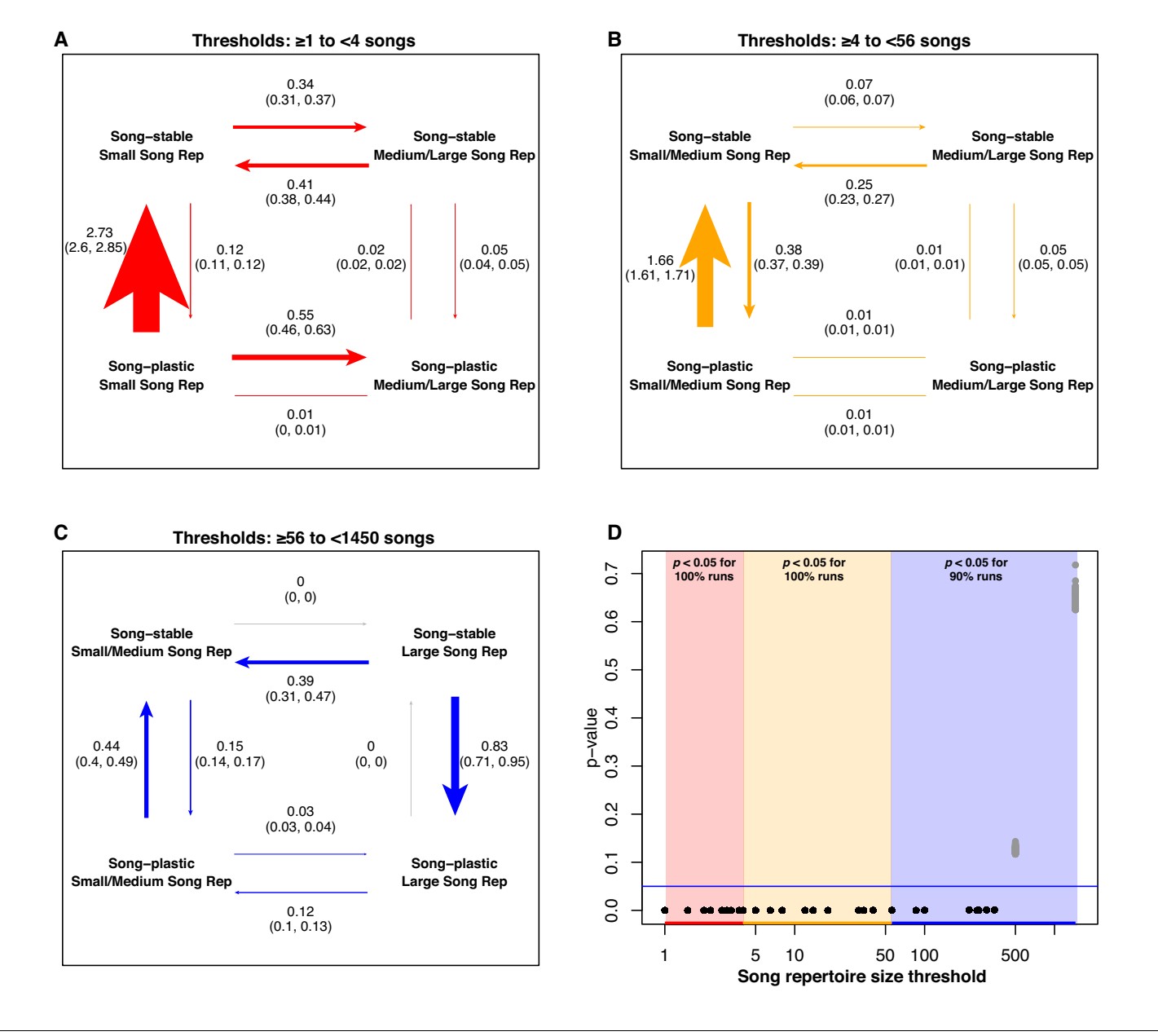

**Figure 7.** Analysis of correlated evolution between adult song stability/plasticity and song repertoire size. Labeling is the same as in *Figure 6*.
DOI: https://doi.org/10.7554/eLife.44454.036

The following source data is available for figure 7:

**Source data 1.** BayesTraits analysis of song repertoire and learning window, jackknifed across families.
DOI: https://doi.org/10.7554/eLife.44454.037

Our findings shed new light on the broader subject of song evolution, specifically the evolution of the process of song learning. We hypothesized that sexual selection on certain aspects of song could in turn alter the selection pressures on the length of the song-learning window. Here, we performed phylogenetically controlled analyses across 67 songbird species to assess the evolutionary interactions between song characteristics and song plasticity in adulthood. With these analyses, we show the first evidence for this evolutionary relationship. Adult song stability versus plasticity may be evolutionarily dependent upon the properties of the song itself: large syllable and song repertoires

appear to drive the evolution of adult song plasticity and thus open-ended song learning. This provides context for the remarkable interspecific variation in song-learning windows across the songbird lineage and suggests an evolutionary mechanism by which sexual selection might have influenced the evolution of songbird brains.

## Materials and methods

### Data collection

It is difficult to precisely measure the length of the song-learning window in both field and lab studies. In field studies, if a bird is recorded singing a new song element in its second year, researchers often cannot rule out the possibility that it learned that element during its first year but did not incorporate it into its song until later (*Marler and Peters, 1981*; *Vargas-Castro et al., 2015*). Likewise, if a bird learns a new element in its second year but elects not to produce it, a recordist would be unlikely to capture it. In contrast, raising birds in the laboratory with known song exposure enables researchers to assess when a bird learned a particular song element (*Chaiken et al., 1994*; *Marler and Peters, 1988*; *Nelson (1998)*; *Nottebohm, 1969*) but also raises questions about whether the lab-reared birds are exhibiting their normal behaviors (*Baptista and Petrinovich, 1984*; *Kroodsma and Pickert, 1984*). In addition, these lab-rearing procedures have only been performed in a handful of species. In this paper, we examined studies that include longitudinal measures of adult song stability versus plasticity (e.g. *Nordby et al., 2002*) to determine whether song is modified across time as a proxy for the length of the song learning window.

We performed a literature search to gather information on the stability or plasticity of male song in oscine species. Studies with information about learning style were found via Google Scholar using the following search terms: [species name] or [common name] in combination with 'open-ended,' 'close-ended,' 'closed-ended,' 'age-limited,' 'crystal*,'' 'adult learning,' and 'song changes.' We used three strategies to assign the song stability classification for a species. We first defined a species as having adult song stability ('song-stable species') if the literature indicated that males did not modify their songs after the first breeding season. Species in which males modified their syllable repertoires after their first breeding season were classified as having adult song plasticity ('song-plastic species'). This strategy was meant to approximate the dichotomy of open- and closed-ended learners often used in the birdsong field. We made two exceptions to this for the species *Cacicus cela* and *Phoenicurus ochruros*, in which males do not gain their mature plumage until their second breeding season and may therefore be delayed in reaching sexual maturity relative to other bird species (*Draganoiu et al., 2014*; *Trainer and Parsons, 2002*). Because these birds cease modifying their repertoires before reaching their second breeding season with mature plumage, they were considered song-stable (*Source data 1*). Additionally, past research defined *Melospiza lincolnii* as an 'open-ended improviser.' It is unclear whether improvisation throughout the lifespan is equivalent to learning throughout the lifespan, however it does fit our definition for adult song plasticity. For the main analysis, we considered this species to have adult song plasticity, but we repeated the main analysis with this species reclassified as having adult song stability. This reclassification produced a negligible effect on our results (*Supplementary file 1* Tables S19-20). For our second strategy, we separated song-stable species into two sub-groups: early song-stable (species that cease modifying their songs before the first breeding season) and delayed song-stable (species that modify their songs during their first breeding season but not after). There was not enough information to make this determination for some species, so our dataset was reduced to 59 species. We used these same 59 species to generate a continuous measure of song stability for our third strategy. Because no information was available about the prevalence of song changes beyond the second breeding season for most of the species in our dataset, this measure only ranges from 0 to 2 years, and all song-plastic species were assigned a value of 2. Furthermore, it was not clear exactly when most of the delayed song-stable species stopped learning, so they were given a value of 1.33, at which point the first breeding season should have ended. The two species mentioned above that display delayed song and plumage maturation were assigned a value of 2.

To gather data on the song characteristics, we performed a literature search via Google Scholar and Web of Science using the search terms 'Passeriformes' or [species name] in combination with 'syllable repertoire' or 'song repertoire.' This yielded a mix of primary sources and studies that had

previously aggregated repertoire size data. We also gathered data from the curated field guides Birds of North America (*Rodewald, 2015*) and Handbook of Birds of the World (*del Hoyo et al., 1992*). We did not perform explicit searches for any of the other included song characteristics, but we collected this data whenever we encountered it. Song characteristic nomenclature is variable across studies, so, when possible, we read the methods of the primary sources to ensure that the data collected were consistent with our definitions for song characteristics.

We utilized the following definitions for song characteristics:

1. Syllable repertoire - the mean number of unique syllables sung per individual.
2. Syllables per song - the mean number of unique syllables per song.
3. Song repertoire - the mean number of unique songs produced per individual.
4. Song duration - the mean length of a song, delineated by pauses or beginnings of a repeated motif (seconds).
5. Intersong interval - the mean pause length between songs (seconds).
6. Song rate - the mean number of songs produced per minute (calculated value; Rate = 60/ (Duration + Interval)).
7. Continuity - the fraction of singing time spent actively producing song (calculated value; Continuity = Duration/(Duration + Interval)).

For three species with song-learning window data, syllable repertoire size estimates were not available in the literature, so we estimated these repertoire sizes from published sources or song recordings. For *Philesturnus rufusater* (*Supplementary file 1* Table S21), (*Jenkins, 1978*) coded songs into different types and gave the song repertoire types of each male studied in *Table 2* of that paper. Information from one male was missing from this table. We deduced the repertoire of the missing male by first looking at *Table 1* from that paper (Jenkins's *Table 1*), which showed that 16 males had a repertoire size of one song. Only 15 of the males in Jenkins's *Table 2* had a repertoire size of one song. Thus, the missing male had a repertoire size of one song. We then compared the bands of males present in Jenkins's *Table 2* to the territory map in *Figure 7* of that paper, and A_RW was the only male missing from Jenkins's *Table 2*. A_RW was located in the DC region of Jenkins's *Figure 7*, so we assigned that as his repertoire. Jenkins notes that neighboring males share song types, so the only other song A_RW could have known instead of DC was ZZ, which has the same number of unique syllables as DC. For *Geospiza fortis* (*Supplementary file 1* Table S22) and *G. scandens* (*Supplementary file 1* Table S23), we used recordings from the Macaulay Library at the Cornell Lab of Ornithology (*Cornell Lab of Ornithology, 2019*) to estimate the syllable repertoire size and syllables per song. For *G. fortis*, we also used the sonogram examples present in *Grant and Grant (1996)*.

When the song repertoire for a species equalled one, we assumed that its syllable repertoire was equal to its number of syllables per song. In many cases, there were multiple studies that gave different estimates for a given song variable in one species. To handle these discrepancies, we created three datasets (*Source data 2*). For one, the main dataset, we used the median value across studies. For the second dataset, we used the minimum value reported in the literature for each species, and for the third, we used the maximum value reported in the literature. We log normalized all song trait data. The three datasets revealed similar results; analyses using the maximum and minimum literature values are presented in *Supplementary file 1* Tables S1-S5.

We also cataloged data on mating behavior at the species level. In particular, we assembled binary classifications of social mating system (monogamy vs polygyny) and extra-pair paternity (low EPP vs high EPP). We considered a species to be monogamous or polygynous when a source unambiguously categorized that species' social mating system; that is, we did not assign a social mating system to species labeled 'probably,' 'usually,' 'mostly,' 'normally,' 'typically,' and 'generally monogamous/polygynous,' etc. unless quantitative measurements were also provided. When quantitative data were available, species were defined as polygynous when at least 5% of males had more than one social mate, as in *Snyder and Creanza (2019)*. A review of extra-pair paternity studies estimated an average of ~11% of offspring per nest were attributable to extra-pair mates across species (*Griffith et al., 2002*). In accordance with this estimate and with previous studies that used a binary classification of EPP (*Snyder and Creanza, 2019*; *Soma and Garamszegi, 2011*), we used a 10% threshold for either extra-pair young or nests containing at least one extra-pair chick to estimate the

frequency of extra-pair paternity in that species (<10% = low EPP; ≥10% = high EPP, *Source data 2*).

## Assessing the evolutionary history of adult song plasticity

To predict the rate of transition between adult song stability and adult song plasticity, we used the `ace` function from the R package `Phytools` and a phylogeny from *Jetz et al. (2012)*. We note that this phylogeny has broad coverage of oscine songbirds but is based on limited genetic data (often mitochondrial), so the topology could change as more avian genomes are sequenced (*Jarvis et al., 2014*; *Lamichhaney et al., 2015*). With this tree, we tested whether an all rates different (ARD) model fit the data significantly better than the equal rates (ER) model using an ANOVA. We then used the better-fit equal rates model to generate 10,000 trees with `make.simmap` (`Phytools`). This function uses the rate from `ace` and a phylogenetic tree with annotated tips to create stochastic simulation maps for the potential evolutionary transitions between the song-stable and song-plastic states. We found the predicted ancestral state for each of these 10,000 simulations and used `countSimmap` (`Phytools`) to count the total number of transitions that occurred in each map. The minimum number of predicted evolutionary transitions across these 10,000 simulations was considered to be the most parsimonious; we also compared this to a manual count of evolutionary transitions starting from either ancestral state.

## Detecting differences in song characteristic evolution in song-stable vs. song-plastic species

To test whether there were significant differences between song-stable species and song-plastic species for the song traits, we performed a phylogenetically controlled ANOVA (`phylANOVA`, `Phytools`) for each song characteristic. We repeated this analysis with the subset of species we classified into early song-stable, delayed song-stable, and song-plastic. Because there were relatively few early song-stable species in this dataset, we only performed this re-analysis for song traits that had data for at least nine early song-stable species (syllable repertoire size: nine species with early song stability, song repertoire size: nine species, and syllables per song: ten species). In this paper, we visualize the predicted ancestral traits on the phylogenetic tree with color and pie graphs, however, the raw values are available in *Figure 1—source data 1*. To test for correlations between song characteristics and the continuous values for the duration of song plasticity, we performed a phylogenetic generalized least squares (PGLS) analysis. We used the function `gls` (R package: `nlme`), with the 'correlation' parameter lambda computed using the function `corPagel` (R package: `ape`).

To test whether adult song stability or plasticity affected the rate of evolution for the song characteristics, we used the function `brownie.lite` (R package: `phytools`). This function first calculates a one-rate model of evolution for a song characteristic using a phylogenetic tree and the current states of the tips for that song trait. This one-rate model assumes that change in the value of the song characteristic is random across evolutionary time and can be approximated by Brownian motion. Next, a model is generated wherein two different rates are calculated; this two-rate model assumes that the evolution of the song characteristic has one rate in the song-stable state and a different rate in the song-plastic state. This model requires estimations for the ancestral states of song stability for each branch of the phylogeny. To create these estimates, we used the function `ace` to calculate the rate of transition between the song-stable and song-plastic states for the full dataset. We then used these transition rates to generate 1300 different stochastic simulation maps (`make.simmap`) for the subset of species that had data for each song trait. For the Brownie analysis, we tested whether the two-rate model fit the data significantly better than the one-rate model by performing a chi-square test on the mean log likelihoods of the two models. We repeated this analysis for the set of species we classified as early song-stable, delayed song-stable, and song-plastic for traits for which we had data on at least nine early song-stable species. We compared the three-rate model to the one-rate model. We also reran the two-rate model in this reduced dataset by combining the early and delayed song-stable groups and testing whether the three-rate model was better than the two-rate model. Because the delayed song-stable trace peaked at a similar position to the song-plastic trace for syllable and song repertoire size (*Figure 4D–E*), we also compared the three-rate model to another version of the two-rate model, in which one group was early song-stable (shorter learning), and the other was delayed song-stable and song-plastic combined (longer learning).

We used BayesTraits to test for correlated evolution between song stability and song characteristics, or, in other words, whether the rate and direction of evolutionary transitions of one trait are dependent on the state of another trait, and whether an order of transition events can be inferred. Specifically, we tested the hypothesis that an evolutionary change in song stability increases the likelihood of an evolutionary change in certain song variables or mating behaviors, or vice versa. BayesTraits compares two models of discrete trait evolution for a pair of binary traits and a given phylogenetic tree: 1) an independent model (i.e. the evolution of one trait does not depend on the other trait) and 2) a dependent model (i.e. the evolutionary transitions of each trait depend on the state of the other trait, suggesting correlated evolution). Using the maximum likelihood method, BayesTraits reports marginal likelihoods for the computed dependent and independent models (function `Discrete` in package `btw [BayesTraitsWrapper]`), allowing us to determine whether the dependent model describes the data significantly better than the independent model. We used function `LRtest` (package: `lmtest`) to perform the likelihood ratio test. Since this model required both input traits to be binary, we classified the continuous song characteristics as binary groups ('low' or 'high') based on a delineating threshold. Instead of choosing the threshold arbitrarily, we used each unique value of the song characteristic data as the threshold and repeated the analysis 100 times at each threshold. This method of using a spectrum of thresholds to delineate the 'low' and 'high' value categories resulted in transition rates that varied dramatically depending on where the threshold was placed. In essence, when the threshold was set as a value in the bottom third of the unique trait values present in the data, the analysis evaluated the rate of transition from low-to-moderate and larger values for a song trait and vice versa. When the threshold was set as a value in the upper range of the unique trait values present in the data, the analysis calculated the rates of transition from higher song trait values to medium-to-low values. To account for this nuance, we binned the threshold data into two to five bins, with three bins as the default: low (bottom third of unique trait values), medium (middle third) and high (top third). We then calculated the mean of each state transition rate in each bin. In addition to the song characteristics, we also analyzed song stability versus social mating system (i.e. social monogamy or polygyny) and rate of EPP. These analyses were performed for 1000 runs each.

## Jackknife analysis

Some families of birds were well represented in our sample, while others were only represented by one or two species. To test whether any well-represented family significantly skewed our results, we removed each family that was represented by four or more species in the full dataset in turn, and repeated the `phylANOVA`, `brownie.lite`, and `BayesTraits` analyses. Jackknife analyses were only performed when significant results were obtained in the main analysis. Thus, all song traits except continuity were tested in the `phylANOVA` and `brownie.lite` jackknife analysis, while only syllable repertoire, song repertoire, and syllables per song were tested in the `BayesTraits` jackknife analysis. Each Brownie analysis was run on 1300 unique stochastic character maps, and each BayesTraits analysis was repeated 20 times. We determined the family of each species based on its classification in the 2017 version of the eBird Clements Integrated Checklist (*Clements, 2007*). The family Locustella was combined with Acrocephalidae, as Acrocephalidae was paraphyletic when Locustella was considered to be a separate family. The Mimidae family alone had a large effect on the syllables per song metric, so we performed another jackknife analysis with `phylANOVA` and `brownie.lite` by removing each mimid species in turn.

## Correction for multiple testing

We used a Holm-Bonferroni correction to control for testing multiple hypotheses with the same data (*Holm, 1979*). This correction is appropriate for data wherein the outcome of one test is likely to be related to the outcome of another test, as would be the case for song characteristics.

## Additional information

### Funding
No external funding was received for this work

## Author contributions
Cristina M Robinson, Kate T Snyder, Conceptualization, Data curation, Software, Formal analysis, Validation, Investigation, Visualization, Methodology, Writing—original draft, Writing—review and editing; Nicole Creanza, Conceptualization, Resources, Data curation, Software, Formal analysis, Supervision, Validation, Investigation, Visualization, Methodology, Writing—original draft, Writing—review and editing

## Author ORCIDs
Nicole Creanza  https://orcid.org/0000-0001-8821-7383

## Decision letter and Author response
Decision letter https://doi.org/10.7554/eLife.44454.050
Author response https://doi.org/10.7554/eLife.44454.051

# Additional files

## Supplementary files
• Source data 1. Adult song plasticity and stability references.
DOI: https://doi.org/10.7554/eLife.44454.038

• Source data 2. Song feature and mating system data and references.
DOI: https://doi.org/10.7554/eLife.44454.039

• Source code 1. Supplemental data and code.
DOI: https://doi.org/10.7554/eLife.44454.040

• Supplementary file 1. Supplemental tables S1-23 and supplemental table legends.
DOI: https://doi.org/10.7554/eLife.44454.041

• Transparent reporting form
DOI: https://doi.org/10.7554/eLife.44454.042

## Data availability
All data are made available as supplementary information provided with this manuscript, and are also provided at https://github.com/CreanzaLab/SongLearningEvolution (copy archived at https://github.com/elifesciences-publications/SongLearningEvolution).

The following datasets were generated:

| Author(s) | Year | Dataset title | Dataset URL | Database and Identifier |
|---|---|---|---|---|
| Robinson CM, Snyder KT, Creanza N | 2019 | Dataset S1: Song stability data and references | https://github.com/CreanzaLab/SongLearningEvolution/blob/master/Datasets/Supplementary%20Dataset%20S1.docx | GitHub, 1ecc644 |
| Robinson CM, Snyder KT, Creanza N | 2019 | Dataset S2: Song feature data and references | https://github.com/CreanzaLab/SongLearningEvolution/blob/master/Datasets/Supplementary%20Dataset%20S2.xlsx | GitHub, 1ecc644 |

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
