## [Decision Letter]

Thank you for submitting your article "Correlated evolution of repertoire size and song plasticity predicts that sexual selection promotes open-ended learning" for consideration by *eLife*. Your article has been reviewed by three peer reviewers, one of whom is a member of our Board of Reviewing Editors, and the evaluation has been overseen by Patricia Wittkopp as the Senior Editor. The following individual involved in review of your submission has agreed to reveal their identity: Ofer Tchernichovski (Reviewer #3).

The reviewers have discussed the reviews with one another and the Reviewing Editor has drafted this decision to help you prepare a revised submission.

The authors provide some of the first evidence that an important and variable (but previously poorly understood) aspect of bird song learning – the song learning window – may be shaped by selection for large syllable and song repertoires. Thus, selection pressures on the song structure itself may be selecting for this variation, whereas, in the past researchers have often thought that environmental variation and breeding season length were leading factors shaping this trait. This is an exceptionally novel study and finding that is likely to spur much future research in this area and change how we think about how this aspect of song learning behavior evolves. Two of the reviewers had more minor comments, but a third had more serious concerns about presentation and data availability. I have summarized their comments (often sharing them verbatim) below.

Essential major revisions:

Please pay special attention to the comments below on:

1) The appropriateness of ancestral state reconstruction.

2) The methods on how birds that learn only into the beginning of their second breeding season were handled.

3) The fact that 10% EPP seems low for considering those species as having EPP.

4) Including example sonograms and links to audios.

5) Being more explicit that the behaviors are a long a continuum.

In addition, any additional evidence that can be added on the attractor models would also be appreciated. We recognized this might be challenging, however.

Additional specific comments: (from individual reviewers, lightly edited, in a combined list below)

1) The most critical reviewer writes: It is obvious that song learning takes time: the more syllable types and transitions a bird of a certain species 'happens' to learn, the longer it should take. The authors emphasize this as their principal result, but there is much more in this work than this generic conclusion, which by itself, is not very surprising. It is quite obvious to me that repertoire size should be correlated with open ended learning (adult song plasticity). On the other hand, the results about song evolution rate across categories, and about attractor and repeller states in evolutionary dynamics (Figures 4-5) are much more interesting. The problem is that, without some additional analysis, the case is not strong enough and not even clear enough. Throughout the paper no specific example are presented in a manner that could convince the reader that the evolutionary trajectories are interesting and robust – I think they are, but authors chose not to present any examples that could help me judge.

2) It is surprising that a study about the evolution of song complexity and performance does not show a single sonogram, and does not present even a single example of an evolutionary trajectory. Abstract statistical models with measures that are combined across many species can be used for presenting scientific results, but only after specific 'typical' examples are presented. Specifically, the phylogenetic tree presented in Figure 2 contains several trajectories that look interesting: going either from stability to plasticity, or the other way around over several transitions. Those cases should be analyzed separately to examine if adult plasticity evolved before, during or after changes in features such as syllables per song, etc. Those examples should be presented as sonograms (with attached media) so that readers can assess those changes. One way to do this would be to present at least two examples of each type in the main paper, and then the supplement can show several more. Those specific examples should lead the overall abstract analysis.

3) Evolution of adult song plasticity: I am a bit dissatisfied with the dichotomy between open ended and close ended learners in the input set, which could have affected the number of transitions required by model. Say that we had a continuous variable, such as the lifetime duration of song learning. It seems likely that an ancestor with a narrow song learning window (such as in a zebra finch) could gradually evolve an extended period, which only in specific, extreme cases, would show a phenotype of open-ended learning and adult plasticity. So, I would like the authors to look more carefully at the evolutionary trajectories, and within a small number of species assess the possibility of continuity. I am not sure if sufficient data to make such an assessment are available, but I would assume that song development should correlate with sexual maturity and growth rate, which may be easier to compare across species.

4) I would have thought that the rate of song feature changes should be slower when repertoire size is large (as it takes longer to turn a cargo ship than to turn a canoe). So maybe the rate of change analysis should take this issue into account?

5) The analysis of attractor and repeller states is potentially very interesting. But many arbitrary decisions about thresholds and input measures make me worried about this analysis. First of all, there is a need to look at all those results continuously – if there is an attractor state in low threshold and a repeller state in high threshed, it could mean that there is an intermediate state of equilibrium. In other words, the evolutionary trajectories might be 'hovering' around some stable 'prior'. More analysis is needed here across graphs. Here too, I would start by looking at specific trajectories (and please present sonograms).

6) Throughout the paper I kept wondering if there is a simple way of telling which came first: song learning duration or syllable per song? Can you look at specific trajectories to get some clues, even if not statistical?

7) Also, since mimids drive much of the correlations maybe it is a good idea to look at them more carefully as a post hoc analysis. I understand that authors want to rid of unrepresentative example, but this does not mean that the most convincing cases should be ignored…

8) Overall, it would be good to show, at least in the supplement, some examples of evolutionary of complexity and performance side by side with evolution of open-ended learning. I feel that without seeing what's going on in specific cases it is difficult to judge the quality of the data.

9) Regarding data quality of adult song plasticity: authors used a categorical measure (taken from multiple published papers) and there is no mention in the Materials and methods about assessment and quality control of integrating such a categorical assessment across studies. Song traits, on the other hand, are continuous variables. I think that this categorical, binary assessment of song plasticity has limitation that should be discussed.

10) Line numbers would make reviewing easier.

11) It is not clear whether the data included were all from males in each species, or whether some females were included. If a mixture of sexes, I think this factor should controlled for in one or more analyses. The authors mentioned females in the discussion for future studies, but never mentioned (maybe assumed) that they only analyzed studies with males in this work.

12) The authors should make a cautionary statement that simulations may not match the real biology, but approximate it.

13) The Introduction seems excessively long on background material and different hypotheses. I suggest dealing with the various hypothesis in the Results, and only mention two in the Introduction, so that the paper is less repetitive.

14) Also in the Introduction and other parts of the paper, stability and plasticity are treated as different variables (whether in written text or some of the figures). But they are along a continuum, that the authors results support. I suggest making this clearer.

15) On its first mention, the authors need to cite what phylogenetic tree of oscine families that they used. The accuracy of the tree could affect their results. I couldn't find any citation in the main text for the tree used.

16) About the tree results, it looks pretty clear from all the phylogenetic analyses that there was an early split or two splits in the songbird lineage between two predominantly plastic song clades and a predominantly stable song clade. And then some independent deviations from those early splits moving into the other clade's phenotype. Maybe the common songbird ancestor was in the middle of the continuum. This exciting finding of the major clades with plastic vs. stable song should be highlighted in the Abstract and Results.

17) Subsection “Do our results depend on the particular values or families included in analyses?”, first paragraph. I thought other song production factors, like song intervals, were not affected in the above analyses? But they are here, when using maximum or minimum values?

18) The Discussion is on the long side. It could be shortened without losing content, by not repeating the results, but simply moving into a discussion of them.

19) The following two sentences appear to contradict each other: "We did not find evidence for correlated evolution between song stability and EPP. Most of our simulations testing for correlated evolution between mating system and learning window were significant;"

20) Discussion, seventh paragraph. Explain what is HVC, to scientists unfamiliar with the avian brain.

21) Figure 1: Label y axes (log scale). Mention stat results in the legend

22) Figure 2. Legend text is hard to follow, in terms of grammar.

23) Please consider archiving the data in a proper repository so that they can be more readily built upon.

24) Abstract: Song-plastic species. This is not a term I am used to. Do you mean open ended learners? If so, can you use that term? Or at least consider "species with plastic song" – and perhaps briefly define what you mean.

25) Abstract: "Notably…": What is the evidence for this? Is this derived from the above result, or is it from other results. Please preface this with what kind of analysis demonstrated this result.

26) Introduction – "small-scale analysis of comparative evolution": more appropriate = "small-scale comparative analysis".

27) Introduction: "female choice can favor certain characteristics" – can you give a bit more description of process, or what is selected for (similar to the description for neuroplasticity). This is crucial to your findings.

28) Introduction, third paragraph – I would expect this before talking about song-learning windows – as it is the more standard dogma under which bird song is studied.

29) Results – first paragraph: this is all good info, but does not present any results. It seems better suited for the Materials and methods (and, in fact, this useful explanation of why you went with the novel, broad "song-plastic/stable" terminology is left out of the Materials and methods. I would recommend moving more of this text there and skip straight to your classification and Results (with a brief definition here)).

30) Results – second paragraph: the idea of examining the ancestral state of all songbirds for song plasticity is interesting, however, you don't really have an appropriate dataset for this. Because you have so many missing taxa (only 64 species out of ~5000 categorized) and they fall throughout the songbirds (you could predict the ancestral state for a particular family if you had sampled many representatives of that family, or all songbirds if you densely sampled representatives of lineages branching more directly from the common ancestor of all songbirds). However, your dataset is a good one for examining correlated evolution, as you have done, and you can also get an idea of transition rates and when/ how often transitions between song stability and plasticity occur (although, it is good to be aware that this may be a fairly rough estimate just because there are probably many species that are missing among the species sampled). I would consider removing/avoiding mention of ancestral states, and approach this in light of major transitions among these traits. You could mention that the ancestral state for these lineages could not be resolved.

31) Results – paragraph beginning "Unexpectedly" – I'm not sure this is that unexpected (I'm not certain what I would predict here). Perhaps skip this word.

32) Results – subheading "Does song stability or plasticity affect the direction of evolution for song traits and mating system?": seems to me the predicted direction of causality is backwards here. I would predict that song learning strategy is dictated by ecological traits or the characteristics of the songs that are being selected (as you find). Therefore, I suggest alternate subheading and approaching this discussion from the perspective of "Is song stability or plasticity influenced by the evolution of song traits or mating system?"

I think this is also approached in the same way in the Abstract/Introduction. There I also suggest pitching this the opposite way (that's how I automatically think of it).

33) Discussion – first paragraph: "in concert with sexually selected song features" – we don't know that these features are strictly sexually selected. Consider "song features often thought of as sexually selected" or "song features associated with sexual selection".

34) Discussion – third paragraph: the two sentences after "We observed that with song stability….". Can this be explained more simply? A lot of jargon here (repeller/attractor states). Can you explain these trends in terms of what is evolutionarily stable/unstable – e.g. song stability coupled with small syllables and song repertoires is an evolutionarily unstable state, so transitions away from it are high.

35) Discussion – fourth paragraph: I think you undersell the importance of mimics (not strictly mimids) in songbird song evolution here, and the implications of your result including the mimids. Mimicry is common throughout songbirds – including in the earliest-branching songbird lineage (the lyrebirds). Really the sample size of the study is small compared to all songbirds (understandably limited by our knowledge of the song learning window for most species). However, I think it would be nice to acknowledge here or elsewhere that this is a relatively small sample, but that mimicry is common in songbirds, so the pattern seen in the mimids may be meaningful and it will be interesting to see how it holds up as we learn more.

36) Discussion – overall comment – there are a lot of thoughtful points here, but the Discussion is rather long. I wonder if these can be delivered more briefly. Perhaps some points can be combined, or overall length cut?

37) Materials and methods – first paragraph: "adult song plasticity ('song-stable species')" – I think there is a typo here – doesn't this statement refer to species without song plasticity?

38) Materials and methods – definitions for 'Song-stable/plastic': there seems to be a gap in your definitions – which leaves birds that learn prior to their second breeding season but do not subsequently modify their song following the second breeding season. I am under the impression that this is not an uncommon strategy. How did you handle these species?

39) Materials and methods: defined high EPP when at least 10% of offspring had a genetic male parent other than social male. This seems low to me. What was the justification for this cutoff?

---

## [Author Response]

Essential major revisions:Please pay special attention to the comments below on:1) The appropriateness of ancestral state reconstruction.2) The methods on how birds that learn only into the beginning of their second breeding season were handled.3) The fact that 10% EPP seems low for considering those species as having EPP.4) Including example sonograms and links to audios.5) Being more explicit that the behaviors are a long a continuum.In addition, any additional evidence that can be added on the attractor models would also be appreciated. We recognized this might be challenging, however.Additional specific comments: (from individual reviewers, lightly edited, in a combined list below)1) The most critical reviewer writes: It is obvious that song learning takes time: the more syllable types and transitions a bird of a certain species 'happens' to learn, the longer it should take. The authors emphasize this as their principal result, but there is much more in this work than this generic conclusion, which by itself, is not very surprising. It is quite obvious to me that repertoire size should be correlated with open ended learning (adult song plasticity). On the other hand, the results about song evolution rate across categories, and about attractor and repeller states in evolutionary dynamics (Figures 4-5) are much more interesting. The problem is that, without some additional analysis, the case is not strong enough and not even clear enough. Throughout the paper no specific example are presented in a manner that could convince the reader that the evolutionary trajectories are interesting and robust – I think they are, but authors chose not to present any examples that could help me judge.

Thank you for this feedback! We agree that it is appealingly intuitive that larger repertoires would be correlated with adult song plasticity, but there is of course nuance in this relationship:

We note that (1) the distribution of repertoire sizes shows substantial overlap between species with adult song plasticity and adult song stability (Figure 1), (2) that there are species that do not conform to this trend (*Acrocephalus palustris, Philesturnus rufusater,* etc., Figure 2), and (3) that several species with adult song plasticity have been shown to modify which syllables they perform as they age rather than significantly increasing their repertoire sizes over time. Thus, it was important to establish the evolutionary correlation between larger repertoires and longer periods of song plasticity as the baseline for the rest of our study. Your point is well taken that, in an effort to conduct a broad evolutionary study, we did not highlight any specific examples or show individual songs. We now highlight in Figure 3 a genus that appears to show a relatively recent evolutionary trajectory that includes both a transition to closedended learning and a corresponding decrease in repertoire size. Since our repertoire estimates were primarily pulled from the literature (except in a few cases where we note that we estimated repertoire size ourselves from song data), we do not want to mistakenly give the impression that we routinely analyzed individual song recordings for our analysis, and we hope we balanced this appropriately.

2) It is surprising that a study about the evolution of song complexity and performance does not show a single sonogram, and does not present even a single example of an evolutionary trajectory. Abstract statistical models with measures that are combined across many species can be used for presenting scientific results, but only after specific 'typical' examples are presented. Specifically, the phylogenetic tree presented in Figure 2 contains several trajectories that look interesting: going either from stability to plasticity, or the other way around over several transitions. Those cases should be analyzed separately to examine if adult plasticity evolved before, during or after changes in features such as syllables per song, etc. Those examples should be presented as sonograms (with attached media) so that readers can assess those changes. One way to do this would be to present at least two examples of each type in the main paper, and then the supplement can show several more. Those specific examples should lead the overall abstract analysis.

We originally did not include any sonograms because we gathered our values regarding song complexity and performance from specieslevel estimates from previously published works. As mentioned above, we have now highlighted one example of song stability changing within a genus and included sonograms from xenocanto for visual inspection.

As far as analyzing some of these interesting cases separately to test for order of evolution, we now clarify in the text that this was the hypothesis we were testing with the Bayes Traits analysis. This test for correlated evolution can sometimes reveal a predominant order of transitional events, though it did not conclusively do so in our study. We did find that a transition in learning window was often more likely than a transition in repertoire size when comparing the predicted rate of evolutionary transitions out of unstable states, and we posit in the text that this change in learning window might often come first.

Since you specifically mentioned syllables per song, we also want to clarify those results: since all of the studied mimids were song plastic, and the change in the rate of evolution depended almost entirely on one species (*Mimus gilvus*), we would hypothesize that this change in syllables per song occurred after the transition to songplasticity, though more data on mimids would be needed to be sure.

3) Evolution of adult song plasticity: I am a bit dissatisfied with the dichotomy between open ended and close ended learners in the input set, which could have affected the number of transitions required by model. Say that we had a continuous variable, such as the lifetime duration of song learning. It seems likely that an ancestor with a narrow song learning window (such as in a zebra finch) could gradually evolve an extended period, which only in specific, extreme cases, would show a phenotype of open-ended learning and adult plasticity. So, I would like the authors to look more carefully at the evolutionary trajectories, and within a small number of species assess the possibility of continuity. I am not sure if sufficient data to make such an assessment are available, but I would assume that song development should correlate with sexual maturity and growth rate, which may be easier to compare across species.

Thank you for this helpful feedback. In response, we have implemented two new sets of analyses that we think have strengthened the manuscript.

(As a side note: for this revision, we were able to add a few more species to our analysis. We had the great honor of hosting Rosemary and Peter Grant for our studentinvited seminar series, and cofirstauthors Cristina and Kate described this project to them. They were excited to see our results, and they also helped us add two of Darwin’s finches to our dataset. We were also able to determine repertoire size data for another species, bringing our total to 67 species.)

To address the gap implied by our original framing, we modified our definition of songplastic to be birds that modify their repertoires after their first breeding season. We then revisited the literature on each of our species to gather more detailed information about the length of the songlearning window if available.

First, when possible, we reclassified the species into three categories: 1) those that stop changing their song before their first breeding season (early songstable), 2) those that modify their songs during their first breeding season but not after (delayed songstable), and 3) those that modify their songs after their first breeding season is over (songplastic). Species that showed delayed learning until their second year and also delayed plumage maturation were added to the delayed songstable group. This threestate classification was possible for 59 out of the 67 species.

Second, for the same 59 species, we used reported estimates of the ages at which song stabilizes in a species to create a continuous measure of the songlearning window. There were exceedingly few studies that examined song changes after the second breeding season, so our continuous metric ranges from 0 to 2 years.

Thus, we can now analyze the phenomenon of song stability/plasticity in three ways; in addition to the previous binary categorization (adult song stability vs. adult song plasticity) we have classified the data as a threestate discrete trait and as a continuous trait.

We present results using these two new classification schemes in the main text. As described in our response to the previous comment, we also highlight a specific example with spectrograms to illustrate interesting evolutionary transitions.

4) I would have thought that the rate of song feature changes should be slower when repertoire size is large (as it takes longer to turn a cargo ship than to turn a canoe). So maybe the rate of change analysis should take this issue into account?

We gave a lot of thought to this question. (A limitation, of course, is that archived recordings do not go back far enough to truly know the rate of song change on an evolutionary scale.) One implication of the canoe/cargo ship metaphor is that species with large and small repertoires should have a similar scale of variation in song traits: for example, if, on average, syllables repertoires vary by a few syllable per generation, then a small repertoire would indeed change faster than a large repertoire. We analyzed data across species to see whether this metaphor holds.

We plotted mean repertoire size versus standard deviation in repertoire size (data from Garamszegi et al., 2005).

From this plot, we notice that the standard deviation of syllable repertoire size increases with the mean syllable repertoire size. Thus, it appears that a cargo ship might be able to change at the same rate as a canoe, so to speak. We might hypothesize that other song features would show a similar pattern: the standard deviation of song duration, rate, etc. might also increase as syllable repertoire size increases.

5) The analysis of attractor and repeller states is potentially very interesting. But many arbitrary decisions about thresholds and input measures make me worried about this analysis. First of all, there is a need to look at all those results continuously – if there is an attractor state in low threshold and a repeller state in high threshed, it could mean that there is an intermediate state of equilibrium. In other words, the evolutionary trajectories might be 'hovering' around some stable 'prior'. More analysis is needed here across graphs. Here too, I would start by looking at specific trajectories (and please present sonograms).

Thank you we have addressed this comment in several ways. First, we have recoded the songwindow data as a continuous metric (amount of time that the song is plastic, described above), and performed a PGLS (Phylogenetic Generalized Least Squares regression) test to detect evolutionary correlations in continuous characters. For syllable repertoire and song repertoire, we found a significant correlation between the length of the learning window and repertoire size when controlling for phylogenetic relatedness. We also plotted the continuous data in Figure 1D, so that the reader can evaluate it. This figure illustrates that learning windows of less than one year are primarily associated with repertoires of 10 or fewer syllables; repertoire sizes around one year are compatible with a range of repertoire sizes below ~50 syllables, and learning during and after the second breeding season corresponds to a wide range of possible repertoire sizes, mostly between ~20 and ~500 syllables.

We also examined whether the decision to break species into 3 bins was influencing our results by also dividing the data into 2, 4, and 5 bins. We saw similar results regardless of the number of bins used, where the smallest repertoire sizes with song plasticity were evolutionarily unstable, and the largest repertoires with song stability were evolutionarily unstable. For bins in which the threshold was set in the intermediate range, we also saw similar patterns for syllable repertoire, though it does appear that in the range of ~1460 syllables, there is little evolution between song stability states. Perhaps this range represents an equilibrium point where birds can evolve their repertoire sizes independently of their song stability. This concurs with the results of the continuous analysis noted above: it would make sense that we do not see a high rate of evolution of the length of the learning window in this threshold range, since repertoires of ~1460 syllables are common in both species with learning windows lasting around one year (delayed stability) and those with learning windows lasting two or more years (song plasticity) These results suggest that the evolutionary transitions in repertoire size within this range does not necessarily require alteration of the songlearning window length. For song repertoire, the intermediate bins show that species with smaller song repertoires tend to transition from song plasticity to song stability but there is little change in song repertoire sizes.

We also conducted another test (Author response image 2) and can include in the main text if desired.

Regarding the possibility of an equilibrium state, we made note of an interesting change in repertoire size stability when we repeated the BayesTraits analyses with a different grouping, separating species with early song stability (shorter learning windows) from those with either delayed song stability or adult song plasticity (longer learning windows). In these analyses, we observed that the rate of transition between shorter and longer learning windows was reduced

compared to when species were grouped as songstable and songplastic, hinting that the predominant evolutionary transitions might be between species with delayed song stability and adult song plasticity. The “longer learning” grouping helped stabilize species with the smallest repertoires and longer leaning by reducing the transition to early song stable. Thus, we hypothesize that delayed song stability can allow species with small repertoires to maintain some song flexibility, but learning extending across multiple years is unstable.

**Author response image 2. respfig2:** 

6) Throughout the paper I kept wondering if there is a simple way of telling which came first: song learning duration or syllable per song? Can you look at specific trajectories to get some clues, even if not statistical?

If there were a consistent pattern of which came first, song learning duration or song complexity, we should see a clear trend with the BayesTraits analysis, which would appear as a single directional arrow away from an unstable combination of traits. Our results were more nuanced: evolutionarily unstable combinations, such as very small repertoires with adult song plasticity or very large repertoires with adult song stability, were able to transition in either trait; one transition did not always precede the other. However, we do report a trend in these rates: the transitions between adult song stability and plasticity appear to occur at a higher rate than transitions in repertoire size, perhaps indicating that the song learning window is more likely to change first, and song complexity changes in response. We now note this trend in the text.

7) Also, since mimids drive much of the correlations maybe it is a good idea to look at them more carefully as a post hoc analysis. I understand that authors want to rid of unrepresentative example, but this does not mean that the most convincing cases should be ignored.

We would like to emphasize that mimids only affected the patterns in syllables per song, not in syllable repertoire size, song repertoire size, or any song performance trait. We note in the text that syllables per song might be a metric that is especially susceptible to differences in listener perception: where does one mockingbird song end and another begin?

We also note that with mimids included, we found an increased rate of evolution of syllables per song, but there was no change in the direction of evolution, and the results regarding the rate of evolution relied on the inclusion of two species in particular, *Mimus gilvus* and *M. polyglottis.* Removing *M. polyglottis* made the rate of evolution nonsignificantly different between songstable and songplastic lineages (*p* = 0.074), while removal of M. gilvus made the difference very nonsignificant (*p* = 0.504). Thus, more research is required before knowing which aspects of this result reflect real biological differences and which aspects hinge on the coding of syllables per song for particular mimids. Furthermore, in our new continuous analysis (PGLS), syllables per song was not significantly correlated with the length of the songlearning window.

In total, this suggests that while mimids warrant further investigation, they are acting differently than other openended learners, and it may be that *M. gilvus* specifically is acting differently from other bird species or that its song is particularly difficult to parse accurately into syllables per song.

8) Overall, it would be good to show, at least in the supplement, some examples of evolutionary of complexity and performance side by side with evolution of open-ended learning. I feel that without seeing what's going on in specific cases it is difficult to judge the quality of the data.

Thank you for this suggestion. First, we apologize if our datasets with all of the specieslevel data were not transmitted properly we fully intended for the reviewers to be able to judge the data on song characteristics and learning window and the sources where we obtained those data. As a precaution, we have uploaded all data and code to github (github.com/CreanzaLab/SongLearningEvolution) in addition to the journal website.

In addition, so that readers can evaluate any specific evolutionary trajectory, we now have a supplemental section in which we provide the estimated ancestral values of both song characteristics and learning window. To do this, we numbered every node in the phylogenies and provide a table with song trait and plasticity/stability estimates for each node (Figure 2—source data 1). Thus, the evolution of complexity and performance is shown side by side with the evolution of openended learning for each of the song characteristics. We think that this complements the colorcoding of the tree branches in the main text, providing a quantitative estimate of each evolutionary trajectory.

9) Regarding data quality of adult song plasticity: authors used a categorical measure (taken from multiple published papers) and there is no mention in the Materials and methods about assessment and quality control of integrating such a categorical assessment across studies. Song traits, on the other hand, are continuous variables. I think that this categorical, binary assessment of song plasticity has limitation that should be discussed.

We have addressed this comment both by explicitly describing our categorization methods in the text and by reanalyzing our data in two new ways, multistate discrete and continuous, as described above.

10) Line numbers would make reviewing easier.

We apologize for this oversight! The current version has line numbers.

11) It is not clear whether the data included were all from males in each species, or whether some females were included. If a mixture of sexes, I think this factor should controlled for in one or more analyses. The authors mentioned females in the discussion for future studies, but never mentioned (maybe assumed) that they only analyzed studies with males in this work.

Thank you – we analyzed male song and song learning without explicitly stating that we did so. We have added text to the Materials and methods to clarify this point.

12) The authors should make a cautionary statement that simulations may not match the real biology, but approximate it.

We now include a sentence to this effect in the Results section.

13) The Introduction seems excessively long on background material and different hypotheses. I suggest dealing with the various hypothesis in the Results, and only mention two in the Introduction, so that the paper is less repetitive.

We have streamlined the Introduction and integrated some of the text into the Results section.

14) Also in the Introduction and other parts of the paper, stability and plasticity are treated as different variables (whether in written text or some of the figures). But they are along a continuum, that the authors results support. I suggest making this clearer.

Please see our third response above.

15) On its first mention, the authors need to cite what phylogenetic tree of oscine families that they used. The accuracy of the tree could affect their results. I couldn't find any citation in the main text for the tree used.

Apologies for this missing citation the tree citation (Jetz et al., 2012) is now mentioned in the Results and Materials and methods sections.

16) About the tree results, it looks pretty clear from all the phylogenetic analyses that there was an early split or two splits in the songbird lineage between two predominantly plastic song clades and a predominantly stable song clade. And then some independent deviations from those early splits moving into the other clade's phenotype. Maybe the common songbird ancestor was in the middle of the continuum. This exciting finding of the major clades with plastic vs. stable song should be highlighted in the Abstract and Results.

Thank you for this suggestion! We now highlight this point in Results and Discussion.

17) Subsection “Do our results depend on the particular values or families included in analyses?”, first paragraph. I thought other song production factors, like song intervals, were not affected in the above analyses? But they are here, when using maximum or minimum values?

Song production/performance characteristics were not significantly different when we compared species with adult song plasticity versus species with adult song stability, but some of these characteristics did appear to evolve significantly faster in songplastic lineages. To be stringent, we repeated all analyses with the minimum and maximum values that we found in the literature to ensure that our results did not hinge on the exact value we used (as in Moore et al., 2011).

18) The Discussion is on the long side. It could be shortened without losing content, by not repeating the results, but simply moving into a discussion of them.

We have revised the Discussion to shorten it, by eliminating repeated results, combining semirepetitive and by streamlining the rest of the text.

19) The following two sentences appear to contradict each other: "We did not findevidence for correlated evolution between song stability and EPP. Most of our simulations testing for correlated evolution between mating system and learning window were significant;"

We apologize for the confusing wording here. We were using “mating system” as a general term for monogamy and polygyny, not including EPP. We have rephrased this section and now refer to mating system as “social mating system” to clarify.

20) Discussion, seventh paragraph. Explain what is HVC, to scientists unfamiliar with the avian brain.

We have clarified the wording of this sentence to define HVC.

21) Figure 1: Label y axes (log scale). Mention stat results in the legend.

Thank you, we have made the suggested changes to the figure and its caption.

22) Figure 2. Legend text is hard to follow, in terms of grammar.

We have revised the caption for this figure.

23) Please consider archiving the data in a proper repository so that they can be more readily built upon.

We apologize that the raw data was not available to the reviewers. For our initial submission of the manuscript, we uploaded the database of information that we collected and analyzed as supplementary tables (Supplementary Datasets S1 and S2), and we uploaded the raw data and code as a zipped archive. We intend for both of these (Supplemental Datasets and Supplemental Code) to be supplemental information that would be available alongside our paper on the journal’s website at publication. In case this supplemental data and code are not provided to reviewers, we have also uploaded them to github.com/CreanzaLab/SongLearningEvolution. In response to this comment, we have also added several files to our supplementary information (e.g. Figure 2—source data 1) so that the raw reconstructed evolutionary states at each node are available in conjunction with the phylogenies.

24) Abstract: Song-plastic species. This is not a term I am used to. Do you mean open ended learners? If so, can you use that term? Or at least consider "species with plastic song" – and perhaps briefly define what you mean.25) Abstract: "Notably…": What is the evidence for this? Is this derived from the above result, or is it from other results. Please preface this with what kind of analysis demonstrated this result.

Thank you for these suggestions; we have rephrased the Abstract accordingly.

26) Introduction – "small-scale analysis of comparative evolution": more appropriate = "small-scale comparative analysis".

We have made this change.

27) Introduction: "female choice can favor certain characteristics" – can you give a bit more description of process, or what is selected for (similar to the description for neuroplasticity). This is crucial to your findings.

We have reworded this section with more details to better parallel the description of neuroplasticity.

28) Introduction, third paragraph – I would expect this before talking about song-learning windows – as it is the more standard dogma under which bird song is studied.

We have edited the Introduction and we hope that it flows better.

29) Results – first paragraph: this is all good information, but does not present any results. It seems better suited for the Materials and methods (and, in fact, this useful explanation of why you went with the novel, broad "song-plastic/stable" terminology is left out of the Materials and methods. I would recommend moving more of this text there and skip straight to your classification and Results (with a brief definition here)).

Thank you! We moved this section to the Materials and methods and instead start the Results with a short definition of our classification systems.

30) Results – second paragraph: the idea of examining the ancestral state of all songbirds for song plasticity is interesting, however, you don't really have an appropriate dataset for this. Because you have so many missing taxa (only 64 species out of ~5000 categorized) and they fall throughout the songbirds (you could predict the ancestral state for a particular family if you had sampled many representatives of that family, or all songbirds if you densely sampled representatives of lineages branching more directly from the common ancestor of all songbirds). However, your dataset is a good one for examining correlated evolution, as you have done, and you can also get an idea of transition rates and when/ how often transitions between song stability and plasticity occur (although, it is good to be aware that this may be a fairly rough estimate just because there are probably many species that are missing among the species sampled). I would consider removing/avoiding mention of ancestral states, and approach this in light of major transitions among these traits. You could mention that the ancestral state for these lineages could not be resolved.

We no longer focus on the ancestral state of all birds other than to mention that we could not resolve it with our current dataset.

31) Results – paragraph beginning "Unexpectedly" – I'm not sure this is that unexpected (I'm not certain what I would predict here). Perhaps skip this word.

We have altered this phrasing.

32) Results – subheading "Does song stability or plasticity affect the direction of evolution for song traits and mating system?": seems to me the predicted direction of causality is backwards here. I would predict that song-learning strategy is dictated by ecological traits or the characteristics of the songs that are being selected (as you find). Therefore, I suggest alternate subheading and approaching this discussion from the perspective of "Is song stability or plasticity influenced by the evolution of song traits or mating system?"I think this is also approached in the same way in the Abstract/Introduction. There I also suggest pitching this the opposite way (that's how I automatically think of it).

We agree and have rephrased the text in the Abstract and Results to reflect this idea.

33) Discussion – first paragraph: "in concert with sexually selected song features" – we don't know that these features are strictly sexually selected. Consider "song features often thought of as sexually selected" or "song features associated with sexual selection".

We have changed this phrasing as suggested; thank you.

34) Discussion – third paragraph: the two sentences after "We observed that with song stability….". Can this be explained more simply? A lot of jargon here (repeller/attractor states). Can you explain these trends in terms of what is evolutionarily stable/unstable – e.g. song stability coupled with small syllables and song repertoires is an evolutionarily unstable state, so transitions away from it are high.

We have rephrased the relevant sections using the suggested terms. When drafting the manuscript, we originally used “evolutionarily stable/unstable,” but then we grew concerned that talking about the evolutionary stability of song stability would become confusing.

35) Discussion – fourth paragraph: I think you undersell the importance of mimics (not strictly mimids) in songbird song evolution here, and the implications of your result including the mimids. Mimicry is common throughout songbirds – including in the earliest-branching songbird lineage (the lyrebirds). Really the sample size of the study is small compared to all songbirds (understandably limited by our knowledge of the song learning window for most species). However, I think it would be nice to acknowledge here or elsewhere that this is a relatively small sample, but that mimicry is common in songbirds, so the pattern seen in the mimids may be meaningful and it will be interesting to see how it holds up as we learn more.

We now address explicitly in the Discussion the interesting and important questions surrounding the evolution of mimicry and its interaction with the songlearning window. To shorten the Discussion, though, we moved some of the previous discussion of our Mimidaespecific findings back to the relevant section in the Results.

36) Discussion – overall comment – there are a lot of thoughtful points here, but the Discussion is rather long. I wonder if these can be delivered more briefly. Perhaps some points can be combined, or overall length cut?

Please see our response to comment #18.

37) Materials and methods – first paragraph: "adult song plasticity ('song-stable species')" – I think there is a typo here – doesn't this statement refer to species without song plasticity?

Thank you for catching this confusing typo! We have corrected this in the text.

38) Materials and methods – definitions for 'Song-stable/plastic': there seems to be a gap in your definitions – which leaves birds that learn prior to their second breeding season but do not subsequently modify their song following the second breeding season. I am under the impression that this is not an uncommon strategy. How did you handle these species?

Please see our response to comment #3.

39) Materials and methods: defined high EPP when at least 10% of offspring had a genetic male parent other than social male. This seems low to me. What was the justification for this cutoff?

We now provide the justification for this cut-off in the text: first, a review of extrapair paternity studies (Griffith et al., 2002) estimated the crossspecies average to be ~11% of offspring per nest to be attributable to extrapair mates, so we wanted ‘low’ EPP to be below average. In addition, previous analyses used a 10% threshold (e.g. Soma and Garamszegi, 2011), and we wanted our study to be in line with previous efforts to delineate low/high EPP.